# Recent Advances in Hydrogel-Based 3D Bioprinting and Its Potential Application in the Treatment of Congenital Heart Disease

**DOI:** 10.3390/biom14070861

**Published:** 2024-07-18

**Authors:** Tasneem Salih, Massimo Caputo, Mohamed T. Ghorbel

**Affiliations:** 1Bristol Heart Institute, Bristol Medical School, University of Bristol, Bristol BS2 8HW, UK; iq18161@bristol.ac.uk (T.S.); m.caputo@bristol.ac.uk (M.C.); 2Cardiac Surgery, University Hospitals Bristol, NHS Foundation Trust, Bristol BS2 8HW, UK

**Keywords:** congenital heart disease, children, 3D bioprinting technologies, biomaterials, grow, hydrogel properties, constructs, cells, biomolecules

## Abstract

Congenital heart disease (CHD) is the most common birth defect, requiring invasive surgery often before a child’s first birthday. Current materials used during CHD surgery lack the ability to grow, remodel, and regenerate. To solve those limitations, 3D bioprinting is an emerging tool with the capability to create tailored constructs based on patients’ own imaging data with the ability to grow and remodel once implanted in children with CHD. It has the potential to integrate multiple bioinks with several cell types and biomolecules within 3D-bioprinted constructs that exhibit good structural fidelity, stability, and mechanical integrity. This review gives an overview of CHD and recent advancements in 3D bioprinting technologies with potential use in the treatment of CHD. Moreover, the selection of appropriate biomaterials based on their chemical, physical, and biological properties that are further manipulated to suit their application are also discussed. An introduction to bioink formulations composed of various biomaterials with emphasis on multiple cell types and biomolecules is briefly overviewed. Vasculogenesis and angiogenesis of prefabricated 3D-bioprinted structures and novel 4D printing technology are also summarized. Finally, we discuss several restrictions and our perspective on future directions in 3D bioprinting technologies in the treatment of CHD.

## 1. Introduction

Congenital heart disease (CHD) is a common type of congenital abnormality present in the cardiovascular structure of infants and can have a major impact on their morbidity, mortality, and quality of life [1]. An estimate of 13.3 million people were globally living with CHD in 2019 with an increase mainly attributed to adolescents and younger adults (15–49 years old; increased by 42%) and middle-aged adults (50–69 years old; increased by 117%) [2]. The incidence of CHD is 12.3 per 1000 live births in the United States [2], 6.9 per 1000 live births in Europe and 9.3 per 1000 live births in Asia [3]. The prevalence of CHD in the United Kingdom was associated with 6.4 per 1000 live births upon which 15% were considered ‘life-threatening’ CHD cases [4]. It is predicted that a minimum of 40,000 infants are expected to have CHD in the United States where 25% of those infants will undergo surgical intervention before finishing their first year [2,3]. The percentage of mortality due to CHD varies from 3% to 7% in industrialized countries to 20% in developing countries [5]. There are three main types of CHD: cyanotic heart disease, left-sided obstruction defects, and septation defects [6,7]. The common types of CHD are bicuspid aortic valve and septation defects [3]. The patients with severe CHD usually undergo corrective heart surgeries to rectify the abnormal cardiovascular tissue, which in turn will decrease the patient’s mortality and morbidity rates and enhance their quality of life [8]. The severe CHD includes coarctation of the aorta and critical aortic stenosis; however, other CHDs such as small muscular ventricular septal defect are minor because the defect resolves itself in early childhood [4]. It has been estimated that 80% of infants born with CHD live until 16 years of age; however, in the long term, they will be exposed to increased risk of cognitive and motor deficits and psychological problems which will have a major negative impact on the quality of life [4,7]. Several types of imaging modalities have been used for the early detection of CHD to allow for proper care at the time of delivery or soon after birth to avoid cardiovascular collapse and potential death outcome [9]. 

### 1.1. Treatments

With regards to surgical intervention, the current life-saving corrective heart surgeries or palliative surgeries include the use of prosthetic replacement grafts or mechanical prosthetic valves for treating severe CHD patients suffering from tricuspid atresia and/or pulmonary atresia, coarctation of the aorta, and tetralogy of Fallot (TOF) [10,11,12]. The disadvantages of such treatments are the limited durability; lack of the potential to grow, remodel, or regenerate and could be the cause of patient’s immunological rejection due to the foreign nature of the materials; risk of thrombosis or bleeding effects due to anticoagulation; calcification; endocarditis; and others [10,12,13,14]. Currently, prosthetic grafts, valves, conduits, and patches are made of those short-lived, degenerative materials and as a result, multiple corrective cardiac surgeries are performed during the patient’s lifetime [13]. The most common, clinically relevant, commercial prosthetic materials used for vascular grafts are Dacron, GORE-TEX^®^ (WL Gore & Associates, Inc., Newark, DE, USA) (polytetrafluoroethylene (PTFE)) [15], mechanical valves [10], decellularized xenograft (usually from porcine and bovine) [16,17], cryopreserved homografts, and autografts (from patient’s own tissues) [17]. Even though those materials are durable, using xenografts and homografts is prone to immunological response [18], collagen degradation, long-term dependency on anticoagulation with a risk of bleeding, no growth or remodelling in vivo, limited availability, and high risk of calcification, especially when using homografts. All in all, this will lead to graft degeneration, thereby allowing multiple surgical interventions [14]. Even though autografts have a reduced risk of immunological response and good hemodynamic profile, their availability is limited and they are susceptible to dilation, thereby allowing valve regurgitation [13,15]. Dacron and GORE-TEX^®^ are readily available; however, they are faced with major challenges especially when used in the extra-cardiac Fontan procedure to treat TOF patients. This includes a high rate of stenosis when Dacron grafts are used and surgical delay in newborns until the inferior vena cava has reached a diameter closer to that of its final adult size due to patient somatic overgrowth when GORE-TEX^®^ conduits are used [12]. Due to the abovementioned limitations, alternative options have come into play which have addressed each of those limitations in a precise manner to eradicate the need for multiple surgical interventions and enhance the patient’s quality of life.

### 1.2. Tissue Engineering Approaches

Tissue engineering is a field designated to the amalgamation of medicine, engineering, and biological sciences. Notably, cardiovascular tissue engineering is related to the creation of biocompatible, nonimmunogenic, and non-thrombogenic cardiovascular grafts, valves, conduits, and patches with the assistance of cells and biomolecules to grow and remodel in vivo and ultimately repair the abnormal cardiac tissue without the need for multiple surgical interventions [19]. When choosing scaffolds tailored to the target tissue, it is important to achieve complete degradation of the scaffold (synthetic or decellularized) into non-toxic substances without leaving any residues in the body, thereby reducing the risk of inflammation and calcification if implanted for a long time. Furthermore, this should be replaced by autologous tissue that is able to grow and repair [20]. This approach also involves the seeding of cells in 3D structures to form living tissue-engineered structures that have unique structural and functional characteristics beneficial for restoration, maintenance, and improvement in the targeted tissue [13,21]. The seeding of cells into 3D tissue-engineered structures fall under two main categories that are in vitro and in vivo modalities, whereas the in situ approach refers to the recruitment of endogenous cells [20]. Various types of cells such as induced pluripotent stem cells [22], foetal stem cells, umbilical cord cells, endothelial cells, endothelial progenitor cells, mesenchymal stem cells, bone marrow-derived progenitor cells, and cardiac stem cells, among others can adhere to the scaffolds. This will generate the extracellular matrix and endothelial lining, grow, proliferate, migrate, and differentiate, thereby allowing for maximum integration into the patient’s heart [13]. Stem cell therapy is a prominent field of therapeutics where the cells are directly injected into the infected or wounded sites, thereby inducing the regeneration of the damaged tissue [13]. The disadvantages of using this technique are the poor cell survival and engraftment [13,23,24] and poor concrete evidence in the literature that proves that true regeneration of the defected heart tissue is the result of cardiac cell therapy. However, some studies show that in adult patients, the therapeutic activity from cardiac cell therapy is linked to the release of paracrine factors post-implantation of the cells and before their senescence [25]. Due to the above-mentioned drawbacks, stem cells are combined with tissue-engineered structures to regenerate damaged heart tissues [22]. In 2020, Kang et al. reported the seeding of mesenchymal stem cells (MSCs) taken from cyanotic congenital heart disease (C-CHD) patients and acyanotic congenital heart disease (A-CHD) patients into collagen patches to repair the right ventricular outflow tract (RVOT) of rats, and as such, a comparison based on pro-angiogenic, anti-apoptotic, and inflammation-modulatory potentials was made [26]. In vivo results show that cytokine-immobilized patches seeded with C-CHD MSCs revealed a preserved morphology, extended cell survival, and improved angiogenesis compared to A-CHD MSCs after 4 weeks of RVOT reconstruction.

Several types of biomaterials are chosen for scaffolding; however, the selection of the scaffold should be tailored to the targeted tissue. Decellularized cardiac valve scaffolds from xenogenic or homogenic sources have their own advantages such as the maintenance of extracellular proteins and, subsequently, the maintenance of mechanical properties and low cost [14]. However, the drawbacks are increased stiffness and risk of triggering an immune response due to incomplete decellularization [18]. Interestingly, it is noteworthy to mention that pre-treatment of the scaffolds with host cells from the patient’s own body and biochemical cues could enhance the scaffold’s biocompatibility, recellularization, and improve the cell survival rates [14]. In 2011, Dohmen et al. reported the implantation of autologous vascular endothelial cells seeded on a cryopreserved pulmonary allograft used for reconstruction of the right ventricular outflow tract during a Ross operation on 11 patients [27]. The results showed that all patients survived with no signs of immunological response, no calcification, and excellent hemodynamic performance. However, it is not a necessity to seed cells into scaffolds since several groups have observed spontaneous recellularization in situ, especially recellularization in senescent animal models [28]. In 2016, Sarikouch et al. reported the in situ implantation of non-seeded decellularized pulmonary valve allografts in human to be used in pulmonary valve replacement in congenital heart disease where host cell repopulation occurred and good functionality and no reoperations were needed after a 10-year follow-up [29]. Many years later, the same group showed that decellularized pulmonary homografts presented superior results compared to the commonly used bovine jugular vein conduits for pulmonary valve replacement in each matched group of 319 patients. Furthermore, they have showed that there was a higher freedom from explantation when decellularized pulmonary homografts were used due to reduced degeneration [30]. In 2022, Gao et al. reported the seeding of bone marrow c-kit+ progenitor cells from patients with congenital heart disease on decellularized human-derived cardiac appendage tissues [31]. Results showed stem cell adhesion, proliferation, and differentiation to cardiomyocyte-like cells without any supplements of inducible factors during a 14-day period with an increased protein level of Gata4 and upregulated gene levels of Gata4, Nkx2.5, and cTnT.

Synthetic, biocompatible, biodegradable, and mechanically strong polymeric scaffolds have also been used in the repair of damaged cardiac tissue, keeping in mind the ability of the scaffold to biomimic the native cardiac microenvironment through the promotion of cell adhesion, differentiation, and deposition of the extracellular matrix. Those include polyglicolyc acid (PGA), poly-L-lactic acid (PLLA), polycaprolactone (PCL [32]), polylactic acid (PCLA), and others. In 2005, Shino’ka et al. reported the first clinical trial of the implantation of tubular scaffolds comprised of woven PGA and ε-caprolactone (50:50) and L-lactide seeded with bone marrow mononuclear cells in 25 children with CHD (through the extra-cardiac Fontan procedure) [33]. Results show that the grafts were growing and behaved like the native vasculature; however, after longer-term follow-up [34], the most recurring complication was graft stenosis. In 2019, Best et al. reported the use of electrospun small-diameter tissue-engineered vascular grafts composed of poly-ε-caprolactone-co-l-lactic acid (PCLA) fibres and their performance was evaluated in both venous and arterial circulations of immunodeficient mice [35], and particularly, such a scaffold was successfully used in their clinical trial to treat congenital heart defects. Results showed that in the mid-term, physiologic graft function was observed, characterized by 100% patency and luminal matching with connecting native vessel in both the venous and arterial circulations where the endothelial cell monolayer, macrophage infiltrate, and extracellular matrix deposition and remodelling were achieved. However, in the long-term follow-up, venous grafts formed viable neo-vessels, but arterial grafts exhibited rupture due to aneurysmal dilation. 

To fabricate scaffolds that are durable; biocompatible; non-immunogenic; non-thrombogenic; flexible; have the potential to grow [36]; remodel in vivo; allow for cell adhesion, proliferation, and production of the extracellular matrix; have good hemodynamic performance; easy to implant and handle; and have the ability to be manipulated through the varying concentration of constituents to match the biomechanical environment of the targeted tissue is impossible to attain with the abovementioned technologies. As such, advanced biofabrication tools such as 3D bioprinting have the potential to address most if not all the above-mentioned characteristics of ideal replacement materials for cardiac tissue engineering. Three-dimensional bioprinting is a biofabrication tool that has the potential to create viable grafts, valves, conduits, and patches composed of stem cells and synthetic and natural biomaterials in a highly controlled fashion that are specifically designed to function in physiological environments [37,38]. 

The main reason behind this review is to shed light on the use of hydrogel-based 3D bioprinting in the treatment of CHD. There are numerous publications in 3D bioprinting in cardiovascular tissue engineering; however, very few have addressed the use of 3D bioprinting in CHD applications. This up-to-date review covers various aspects of the use of 3D bioprinting in the treatment of CHD. It is divided into sections that cover the different 3D bioprinting technologies; the role of hydrogel properties in biofabrication, biomolecules, cell sources, vascularization, and angiogenesis; 4D bioprinting; and finally, future directions towards the use of 3D bioprinting in the treatment of CHD. 

## 2. Introduction to 3D Bioprinting 

Three-dimensional bioprinting is an additive manufacturing technology that has made a tremendous impact in the past 20 years in various fields including tissue engineering, regenerative medicine, pharmaceuticals, and many others. This is primarily due to its ability to combat several challenges experienced by experts in the above-mentioned fields which were not dealt with using conventional fabrication methods. Three-dimensional bioprinting has the unique ability to fabricate anatomically accurate 3D shaped structures using computer-aided design and manufacturing (CAD/CAM) technologies captured from medical images such as magnetic resonance imaging (MRI) and computerized tomography (CT) [39]. The main key players in the field of tissue/organ engineering are cells, biomaterials, and biomolecules as shown [40] in Figure 1. The failure in fabricating tissues and organs with a high resolution and speed of complex 3D microarchitectures composed of precisely located multiple cell types interacting with versatile types of biomaterials using conventional fabrication methods has paved the way for using 3D bioprinting.

There are several 3D bioprinting technologies in the market, where each has its own set of advantages and disadvantages; however, the user must be selective about the type of 3D bioprinting technology that he/she will use to fabricate the target tissue/organ. This is dependent on several factors including, but not limited to, print resolution, print speed and time, controllable size, shape and distribution, cell viability, cell density, rheological properties of the materials, and cost [41]. In order to fabricate functional scaffolds that also biomimic native tissues and organs [42], several requirements such as biocompatibility, optimum biodegradation kinetics, biomechanical properties like native tissue, excellent bioprintability, porosity, biomimicry, cell encapsulation, fast crosslinking and gelation, and structural support must be met [41,43]. Specifically, the interconnectivity of pores is necessary for encapsulated cell survival through the transport of oxygen and nutrients to the designated cell sites. One of the main challenges that is currently facing experts in the field of tissue/organ engineering is the ability to fabricate branched, perfused vascularized networks (at single cell level) within the engineered construct of various sizes and shapes to guarantee long-term viability of the engineered constructs [37]. Furthermore, the automated approach for fabricating reproducible, complex 3D tissue microstructures in various sizes and shapes distinguishes the 3D bioprinting technology from the conventional fabrication methods. However, like any technology, there are limitations associated with 3D bioprinting technology, such as (1) the search for biodegradable and biomimetic printable materials that enable cell attachment and proliferation; (2) multifaceted patterning of heterocellular tissues; and (3) preserving cell viability and long-term functionality post-printing until remodelling and regeneration of the defected tissue/organ is achieved [44]. All in all, 3D bioprinting is an advanced fabrication tool used in tissue engineering to build layer-by-layer biomimetic-shaped 3D structures with the precise deposition of multiple cell types, biomaterials, and biomolecules within a controllably sized tissue construct in a timely manner. In the future, it is likely that 3D bioprinting technology will be directed towards in vivo bioprinting where the cells, biomaterials, and biochemical cues are directly deposited on-site on the patient. For the time being, significant progress has been made in skin bioprinting on wound/burn sites [45], the bioprinting of cardiac patches from patients’ own cells [46,47], and bone bioprinting on calvaria defects; however, this will encompass increasingly complex vascularized tissues/organs [37] bioprinted directly after injury or during surgery using robotic surgical tools and a 3D bioprinter in a high-speed and precise fashion.

### 2.1. Three-Dimensional Bioprinting Technologies

According to the ISO/ASTM 52900 standard, the 3D printing technologies are classified into seven technologies that include binder jetting, material extrusion, material jetting, powder bed fusion, sheet lamination, and VAT photopolymerization [48]. The main 3D bioprinting technologies used in tissue/organ engineering are inkjet-based, extrusion-based, laser-assisted, stereolithography (SLA)-based, and scaffold-free-based bioprinting techniques, as shown below [49,50] in Figure 2. The main differences between those techniques are print speed, resolution, cell viability, and the biological materials used for bioprinting.

#### 2.1.1. Inkjet-Based 3D Bioprinting

Inkjet-based printers are also known as drop-on-demand printers and they are the most common type of printer for biological and non-biological applications [51]. In general, this low-cost, easy-to-use, efficient non-contact technique is ideal for fabricating 2D and 3D structures using low-viscosity picolitre bioink droplets deposited into a substrate matrix with increased speed, resolution, and precision [37,39]. There are several types of inkjet bioprinters but the main types are thermal [52] and piezoelectric actuator [53,54] inkjet technologies. 

Thermal inkjet printers are based on the concept of electrically heating the printhead, thereby producing pulses of pressure that force bioink droplets at picolitre volume (1–100 pI) to flow through the nozzle and onto the substrate matrix. Each droplet can contain 10^4^–30^4^ cells with a diameter of <30 µm, thereby offering high resolution. It has been reported that the localized heating which ranges from 200 to 300 °C does not have a significant effect on the cell viability and stability of biomolecules, especially since the short period of heating (~2 µs) produces a temperature rise of 4–10 °C in the printhead [55]. Even though thermal inkjet printers are readily available, low-cost, and have fast printing speeds, there is significant risk of exposing cells to thermal and mechanical stress, nozzle clogging, low droplet directionality, and size precision [37]. Xu et al. reported the deposition of Chinese Hamster Ovary (CHO) and embryonic motoneuron cells into pre-defined patterned structures using a thermal inkjet bioprinter with a cell viability of more than 90% [56]. Chui et al. reported the use of a thermal inkjet 3D bioprinter to fabricate microvasculature encapsulated by human microvascular endothelial cells and fibrin [57]. Results show that the scaffold maintained a proper shape post-printing and the endothelial cells proliferated and formed a tubular structure after being cultured for 14 days.

Piezoelectric inkjet printers use the piezoelectric actuator to produce acoustic waves in response to applied voltage in the printhead, which in turn breaks the liquid into droplets at regular intervals. As opposed to thermal inkjet printing, the size of droplets and rate of ejection are controlled throughout the process and nozzle clogging is reduced due to the presence of the open-pool nozzle-less ejection system [58]. As a result, cell viability and function are increased due to elimination of shear stress and absence of heat and pressure experienced by the cells at the nozzle tip [37,58]. Compaan et al. reported the piezoelectric inkjet bioprinting of 3D constructs composed of 3D silk fibroin and alginate used as a sacrificial template due to the slow gelation of silk fibroin [54]. NIH 3T3 fibroblasts were able to spread and proliferate with increased metabolic activity 5 weeks post-printing within stable silk fibroin constructs. In 2015, Christensen et al. reported the fabrication of vascular-like cellular structures with both vertical and horizontal bifurcations encapsulated with mouse fibroblast-based alginate bioink using piezoelectric inkjet 3D bioprinting. Results showed a post-printing fibroblast cell viability of 90% after 24 h incubation [59].

Despite the ease of use, low cost, and high resolution of inkjet printing, the major drawbacks of this technique are the narrow selectivity of materials due to the bioprinting of only low-viscosity materials (in liquid form), difficulty in fabricating clinically relevant large-scale engineered constructs, slow printing time, and unstable or weak printed structures that do not withstand physiological environment. 

#### 2.1.2. Extrusion-Based 3D Bioprinting

The extrusion-based bioprinting is based on the extrusion of filaments that consist of biomaterials, cells, or bioink through a nozzle with the help of pneumatic and mechanical (piston- or screw-driven) systems [60]. The extruded filaments are layered in specific patterns forming 3D structures using the computer-controlled movement of the stage in the x, y and z directions [61]. Pneumatic systems have poor control over the flow of hydrogel due to the delayed response in compressed gas volume as opposed to piston-driven systems [19]. Furthermore, screw-driven systems have superior spatial control over pneumatic and piston systems especially since they are more suited for high-viscosity hydrogels; however, there is a significant risk of clogging, thereby reducing cell viability and function. Recently, the emergence of advanced extrusion-based bioprinters based on multiple printheads simultaneously extruding multiple bioinks via multiple nozzles has paved the way for printing tissue constructs with high-complexity biomimicking native tissues. This printing technique is one of the most used techniques (after inkjet printing) in the biological and non-biological fields. The main advantages of extrusion-based bioprinting are the high density of cells/bioinks (10^8^–10^9^ cell/mL), continuous deposition of the bioinks in a controlled manner, fabrication of clinically relevant size and anatomical shape within a realistic timeframe, usage of polymers that exhibit shear-thinning behaviour (most polymers have this characteristic), less intricate bioprinting process, and low cost as opposed to other bioprinting techniques [62]. However, the main disadvantages of extrusion-based bioprinting are the ability to accommodate a large range of viscosities of biomaterials (30 to >6 × 10^7^ mPa·s which is translated into nozzle clogging, thereby reducing the cell viability (40–80% cell viability) [37] and cell function due to enhanced shear stress experienced by the cells during bioprinting [62], and the typical diameter of the nozzle is about 150–300 µm, thereby reducing the printing resolution (over 100 µm) and precision since the spatial resolution is dependent on the diameter of the extruded filaments [62]. 

Gaetani et al. reported the use of extrusion-based bioprinting of alginate and RGD-modified alginate with human cardiac-derived cardiomyocyte progenitor cells (hCMPCs) in a 3D tissue matrix where the hCMPCs experienced high cell viability (92% and 89% on day 1 and day 7, respectively) and kept their cardiac lineage and superior expression of early cardiac transcription factors and the sarcomeric protein TroponinT [63]. Pati et al. reported the use of the extrusion-based bioprinting of a cardiac decellularized extracellular matrix with rat myoblasts in 3D grid structures where the cells experienced greater than 90% cell viability with enhanced cardiac differentiation and improved function in vivo [64]. Asulin et al. reported the one-step 3D bioprinting of electronic heart patches composed of three bioinks: extracellular matrix hydrogel encapsulated with cardiac cells; graphite flakes mixed with liquid polydimethylsiloxane (PDMS) as a conductive material for electrodes; and liquid PDMS as a dielectric material for passivation of the electrodes that is a bioinspired model mimicking the contracting myocardium [47]. This soft, stretchable, and flexible engineered tissue construct can monitor its function through extracellular potentials and produce electrical simulation for pacing. 

All in all, extrusion-based bioprinting provides superior performance in terms of stability of 3D structures with higher cell density compared to inkjet and LIFT bioprinters. Furthermore, the deposition of continuous filaments in a controlled manner of clinically relevant-sized 3D tissue constructs within a realistic timeframe is also a main key player in tissue engineering. The reduced cell viability could be countered by using larger nozzles at lower extrusion pressures; however, this will be compromised by having lower printing resolution and print speed.

#### 2.1.3. Laser-Induced Transfer Forward-Based 3D Bioprinting

Laser-induced transfer forward technology (LIFT) (a member of the laser-assisted bioprinting technologies) is based on the high-intensity, focused laser pulses hitting the donor slide which was initially covered with a laser energy absorbing layer (e.g., gold or titanium), thereby causing local evaporation of the absorption layer [19,37]. Such evaporation generates high gas pressure, therefore forcing the ejection of small droplets of bioink to deposit on the collector slide. The collector slide could be covered with hydrogels to relieve the stress on the falling droplets [62]. 

In 2011, Gaebel et al. reported the fabrication of a cardiac patch composed of randomly seeded human umbilical vein endothelial cells (HUVECs) and human mesenchymal stem cells (hMScs) in a vascular pattern and embedded in polyester urethane urea using LIFT. This approach showed increased vessel formation and capillary density with a significant increase in the amalgamation of human cells into the vessels of the murine vascular system and functional improvement in the infarcted rat hearts [65]. In 2015, the same group reported the bioprinting of a cardiac patch composed of human cardiomyocyte progenitor cells (hCMPCs) in a hyaluronic acid and gelatin matrix, and it was shown that the hCMPCs kept their cardiac phenotype for up to 1 month with a significant reduction in LV remodelling [66]. 

The main advantages of this technology are the absence of clogging due to the presence of non-contact and being nozzle-free, thereby increasing cell viability and function, having a high degree of spatial control and accurate deposition of bioink, and printing high-density cells (10^8^ cells/mL) containing biomaterials with a wide range of viscosities (1–300 mPa·s) at a high speed and resolution (single cell per drop). However, the main disadvantages of this technology are the inability to print clinically relevant hierarchal 3D structures within a realistic time frame [19], the requirement of rapid gelation kinetics to achieve excellent shape fidelity, high cost, contamination of the tissue construct due to the presence of metallic residues from the vaporization of the metallic laser-absorbing layer [37], and complicated laser processes [62]. All in all, LIFT is not the ideal technology to use in widespread tissue engineering applications. 

#### 2.1.4. Stereolithography (SLA)-Based 3D Bioprinting

Stereolithography (a member of the laser-assisted bioprinting technologies) is based on the additive build-up of tissue constructs layer-by-layer from the bottom to the top in a container containing a photopolymer which selectively solidifies when exposed to a digital light projector [62,67,68]. The main advantages include simplicity, a high-speed manufacturing process, and the absence of clogging due to the presence of non-contact and being nozzle-free, thereby reducing shear-related cell death, increasing cell viability (>90%), and having a high resolution (50–100 µm) [69]. However, the limitations include the potential cell DNA damage due to exposure to UV light, restricted range of laser printable materials, and inability to bioprint multiple cells/material scaffolds [67]. In 2018, Izadifar et al. reported the fabrication of a nano-reinforced cardiac patch composed of hybrid hydrogel constructs (methacrylated collagen (MeCol) encapsulated by human coronary artery endothelial cells and carboxyl functionalized carbon nanotubes (CNTs) incorporated in an alginate matrix) using a UV-assisted 3D bioprinting approach [70]. Results showed that the alginate coated CNTs enhanced the electrical and mechanical properties in addition to the enhanced cellular attachment and elongation when compared to native alginate. Furthermore, the hybrid constructs exhibited enhanced stiffness and electrical conductivity within the physiologically relevant frequency range (~5 Hz), in addition to enhanced in vitro cellular proliferation, migration, and differentiation (lumen-like formation) during a period of 10 days of incubation at a specific CNT mass ratio. Melchiorri et al. reported the use of digital light stereolithography in the fabrication of customized polymeric vascular, acellular grafts composed of polypropylene fumarate (PPF) in the treatment of the coarctation of the aorta [71]. Results showed that the customized grafts remained patent for over 6 months with no signs of thrombosis, stenosis, and aneurysm when implanted in vivo in mice. Furthermore, endothelial cells and smooth muscle cells covered the surface of the grafts, suggesting neotissue formation.

#### 2.1.5. Scaffold-Free-Based 3D Bioprinting

Scaffold-free-based 3D bioprinting is based on the deposition of spheroids on a needle array to create tissues without the need of ECM-based materials [32,72]. The design of the spheroidal structures is based on fundamental developmental principles such as embryological or organoid biology. The cells are not able to easily form stable structures especially if structural cohesion is not achieved because of the cells secreting their own extracellular matrix. As such, the cells are initially pre-assembled in clusters (i.e., spheroids) and they are tested as to whether they have secreted their ‘glue’ or not to be able to maintain large-shaped structures [73]. The main advantages of this technique include the absence of clogging due to being nozzle-free, thereby reducing shear-related cell death, and high speed compared to other bioprinting techniques by using ‘building blocks’ of large spheroids. The limitations of this technology include the triggering of the negative self-immune response when the cellular structure is placed in the damaged/degenerate area [74], long pre-printing preparation, and limited scalability since intercellular communication is important and, as such, only smaller cell-heterogenous structures are printed [73]. Furthermore, the cost of this technology ranges from low for cell suspension to high for cell sheets, and it has low printing resolution and low cell viability for cell suspension and high cell viability of cell sheets compared to other 3D bioprinting technologies [75]. Ong et al. reported the fabrication of the scaffold-free 3D-bioprinted cardiac patches composed of various ratios of human induced pluripotent stem cell-derived cardiomyocytes (hiPSC-CMs), fibroblasts (FBs), and endothelial cells (ECs) fused together to form multicellular spheroids [76]. Results showed that the cardiac tissue patches were beating spontaneously after 3D bioprinting, and they exhibited ventricular-like action potential waveforms with electrical conduction uniformity throughout the patches. When placed in vivo in a native rat myocardium, the pre-vascularized cardiac patches exhibited infusion with recipient blood, indicating spontaneous anastomosis. Recently, Kawai et al. reported the fabrication of 3D-bioprinted scaffold-free tubular engineered heart tissues (T-ENTs) made up of human induced pluripotent stem cells derived from cardiomyocytes, human umbilical vein endothelial cells, and human fibroblasts (hiPSCs-COs) in the treatment of congenital heart disease [77]. The T-ENTs were transplanted around the abdominal aorta and inferior vena cava of mice. Results showed that the T-ENTs were beating after 1 min of transplantation in mice and visible striation of the myocardium and vascularization were seen in comparison to the hiPSCs-COs initially transplanted around the aorta or in subcutaneous tissue. 

#### 2.1.6. Other 3D Bioprinting Technologies

Other 3D bioprinting technologies include printing within a suspension bath such as omnidirectional 3D printing and freeform reversible embedding of suspended hydrogel (FRESH) printing. Omnidirectional printing is based on the updated version of extrusion-based printing that is mainly focused on printing microvasculature where 3D-printed structures are printed within a reservoir containing a liquid photo-curable matrix [62,78]. The sacrificial ink is initially extruded at the bottom of the reservoir creating a 2D projection of the structure, thereby creating an empty space due to the X-Y movement of the nozzle that is filled up with lower-viscosity fluid from the top. As a result, a 3D volumetric projection of the 2D design is created prior to the UV-polymerization of the liquid photo-curable matrix around the 3D structure, and finally, the sacrificial bioink and liquid matrix are removed by cooling vacuum, thus creating a hollow 3D structure. The main advantages of this approach include the absence of layer-by-layer deposition of bioink which is a time-consuming process, the ability to print with low-viscosity biomaterials creating structures with high print resolution and fidelity [79], prevention of the collapse of printed structures due to gravity, and the printing of structures in arbitrary, discrete locations. The limitations include inadequate 3D slicing software for omnidirectional printing that is otherwise adapted to the layer-by-layer deposition of printed structures, removal of the solid 3D-printed structures from suspension media without compromising the structural integrity of the 3D-printed structures, and the limited range of deposited bioinks that have printing temperatures that are compatible with the temperatures of the suspension bath [78]. Wu et al. reported the fabrication of a vascular network where sacrificial inks composed of Pluronic F127 diacrylate were omnidirectionally printed in photocurable support gels [80]. The physical properties of the sacrificial inks and photocurable support gels were altered in a way to allow for the deformation of the support gels as the extrusion nozzle moves through it, thereby creating an empty space that was filled up by low-viscosity Pluronic F127 diacrylate. As a result, the printed structure was crosslinked with UV light followed by the removal of sacrificial ink, thereby producing a vasculature network within the tissue construct. To build on this, FRESH printing was developed as a technique by Adam Feinberg’s group to allow for the 3D printing of soft materials within a support bath by depending on the mechanical stability of gelatin microparticles at a precise temperature and pH range, which allows for the crosslinking of collagen [81]. As opposed to the omnidirectional printing, the extrusion nozzle deposits highly acidic collagen bioink as it moves through the pH-neutral-based gelatin support bath, which acts as a Bingham plastic at room temperature without compromising its structural integrity. The deposited collagen bioink undergoes rapid crosslinking within the pH-neutral-based gelatin support bath resulting in a solid, printed structure that is released once the printed construct and support bath are warmed up to 37 °C, thereby melting the gelatin support bath and releasing the printing structure whilst maintaining its structural fidelity. Afterwards, the same group updated their technique to FRESH v2.0 and reported the use of regular, smaller-diameter sized gelatin microparticles (~65 μm to 25 μm), which allowed for the decrease in the diameters of the printed collagen filaments (200 μm to 20 μm) and the presence of a porous microstructure, which led to the promotion of cell infiltration and micro-vascularization [82]. This led to the fabrication of perfusable, multiscale vascular networks, tri-leaflet valves, LAD branching, and a full-sized neonatal-scale human heart using the same technique. Furthermore, functional 3D-bioprinted cardiac ventricles encapsulated by human cardiomyocytes remained viable for up to 28 days and showed directional wave propagation during synchronized contractions with wall thickening up to 14% during peak systole. The same group reported the fabrication of a 3D-bioprinted perfusable contractile heart tube composed of human stem-cell-derived cardiomyocytes and cardiac fibroblasts which remained viable for up to 1 month [83]. Synchronous contractions were exhibited 3–4 days post-printing in addition to anisotropic calcium wave propagation with a conduction velocity of ~5 cm s^−1^. Furthermore, immunofluorescent staining showed the presence of large, interconnected networks of sarcomeric alpha actinin-positive cardiomyocytes. Dvir’s group also reported the use of the FRESH technique to 3D bioprint thick, vascularized, and perfusable cardiac patches composed of cells that were taken directly from the omental tissue of the patient and reprogrammed to pluripotent stem cells and later on differentiated to cardiomyocytes and endothelial cells, and the extracellular matrix was processed into a personalized hydrogel [46]. The personalized, functional, vascularized cardiac patches showed elongated morphology of the cardiomyocytes and striation of actinin and signal propagation above 10 cm s^−1^. 

All in all, each 3D bioprinting technology mentioned above has its own set of advantages and disadvantages. As such, it is imperative to choose the 3D bioprinting technology that best suits the type of target tissue/organ. However, this selection must be aligned with the chosen type of biomaterial, cells, and biomolecules. Each of the main three players must be compatible with each other or else failure in 3D bioprinting a tissue-engineered structure that closely biomimics the target tissue/organ will be achieved. Therefore, the following sections discuss in-depth the role of each main player in the fabrication of tissue-engineered structures. 

## 3. The Role of Hydrogel Properties in Biofabrication

Hydrogel properties play a significant role in determining the type of biofabrication method and cell culture environment needed to generate structural and functional tissue-engineered products that behave like the native tissues [84]. This is determined by their physical, chemical, and biological properties, but most importantly, the printability of hydrogels is dependent on their rheological and crosslinking mechanisms [19]. This is true especially in terms of viscosity since different bioprinting techniques (as shown in the above section) use hydrogels with a limited range of viscosity, such as the inkjet bioprinting technique that uses low-viscosity fluids whilst the extrusion-based bioprinting technique uses high-viscosity hydrogels [19,37]. The type of nozzle used for each biofabrication method is crucial due to the exposure of cells to shear stress which can be detrimental to cells if the shear stress is high, and the duration of fabrication is also a key factor when bioprinting clinically relevant-sized tissue structures [85]. The gelation kinetics should also be taken into consideration especially since the aim is to bioprint tissue-engineered products that have good shape fidelity and mechanical stability within a realistic timeframe [19,37]. Additionally, the biodegradation mechanism and biodegradation kinetics of the hydrogels turning into harmless waste products within an optimum timeframe are also key factors. 

The term ‘bioink’ refers to cell-laden hydrogels that are aqueous and biomimic the natural extracellular matrix which provides cells with an almost ideal supportive environment for survival, growth, migration, and proliferation [19,37]. The choice of hydrogels is dependent on the field of tissue engineering application, and in our case, it is cardiovascular tissue engineering. Finally, the aim is to design cardiac tissue-engineered products that look and function like native cardiac tissues to allow the neighbouring cells to regenerate diseased or damaged cardiac tissues. Most importantly, to achieve the above target, the biological and biomechanical characteristics of hydrogels must be carefully studied, and according to that, a decision is made with regard to the most suited bioprinting technique. 

### 3.1. Rheological Properties

Rheology is the study of the flow of matter due to the application of external force [19]. This property is dependent on several characteristics of the hydrogel but most importantly viscosity, shear thinning behaviour, and yield stress [85]. This is crucial in terms of whether the hydrogel is bioprintable or not, and if true, then a decision is made with regards to the selection of the appropriate bioprinting technique. Viscosity is the resistance of the flow of fluid due to the application of stress, and the key influencing factors on viscosity are polymer concentration, molecular weight of polymeric solutions, shear rate, temperature, and others [19,37,44]. It is important to have optimum polymer concentration and molecular weight since a high polymer concentration is beneficial in terms of good/excellent shape fidelity of the bioprinted 3D tissue constructs; however, it could be detrimental to the encapsulated cells, causing them to experience changes in morphology, metabolic activity, reduction in cell proliferation or migration, or even die as a result of exposure to high shear stresses [86]. With regards to low polymer concentration, the viscosity is reduced, and as a result, surface-tension-driven droplets are formed rather than extruded filaments. Therefore, it is best to have hydrogels with a low concentration of high-molecular-weight polymer to achieve good bioprintability without sacrificing cell survival, proliferation, and migration. 

Shear thinning behaviour is referred to as non-Newtonian behaviour where the viscosity increases as the shear rate decreases [19,37,87]. This is a specifically useful feature to have in extrusion-based bioprinting since a decrease in viscosity at increased shear rates is witnessed as the bioink is flowing smoothly through the nozzle/orifice, thereby reducing the friction forces between the nozzle and cell-laden hydrogel leading to enhanced cell survival. Finally, a sharp increase in viscosity upon deposition of the filaments is achieved, thereby enhancing the shape fidelity and mechanical strength of the bioprinted tissue constructs [86]. Yield stress is another important property that must be taken into consideration when choosing the optimum hydrogel for biofabrication [19,87]. This is defined as the stress required to be surpassed for the initiation of flow. By surpassing the yield stress, the hydrogel will be able to flow and deposit into structures that do not collapse, thereby maintaining their structural integrity and shape fidelity. All in all, ideally, the cell-free hydrogels should have the same/enhanced rheological properties compared with cell-laden hydrogels, as mentioned by Cheng et al. [88]. This means that the encapsulated cells have secreted their own extracellular matrix which added further mechanical strength to the bioprinted tissue constructs.

### 3.2. Crosslinking and Gelation 

The gelation of hydrogel is of utmost importance due to its dependency on whether the printed hydrogel structures can maintain shape fidelity post-printing or not. The type and kinetics of gelation are dependent on the choice of hydrogel [85,89]. Crosslinking is defined by the entanglements of the polymeric chains that are achieved through physical, chemical methods, or the combination of gel-like materials. The crosslinking mechanisms used in biofabrication are physical, chemical, and enzymatic methods [89]. The most prominent physical crosslinking mechanism is the ionic crosslinking, and it has two major advantages that are the excellent biocompatibility with biological products and good gelation kinetics; however, the constructs are mechanically weak, thereby the implementation of an additional crosslinking agent is a necessity [41]. Chemical crosslinking is the association of polymeric chains with covalent bonding yielding constructs with excellent mechanical strength and stability compared with physically crosslinked constructs. This is achieved either through the exposure of cell-laden constructs to chemical crosslinking agents or the formation of photoreactive species upon photoirradiation. The disadvantage of having excellent biomechanical characteristics of bioprinted tissue constructs is the difficulty experienced by biomolecules to diffuse into the tissue as a result of strong interactions between the covalently bonded species [89]. Photopolymerization is one of the most popular crosslinking mechanisms and it is defined as the production of a crosslinked network of low-molecular-weight monomers or oligomers when exposed to radiation [41,90]. The main advantage of using a photo-crosslinking mechanism as opposed to physical crosslinking is the production of mechanically strong and stable constructs. The main disadvantages of this mechanism are the probable senescence of encapsulated cells when irradiated for a long period of time and the high reaction rates required to complete the polymerization process between the nozzle and deposition. Enzymatic crosslinking is a popular mechanism used in tissue engineering specifically for crosslinking fibrin, in which thrombin (a serine protease) is used to convert fibrinogen into fibrin. 

### 3.3. Bioprintability

The major aspect of fabricating tissue engineering constructs is whether the individual components are bioprintable or not [91]. There is no need to search for the ideal bioink that has excellent biocompatibility, biodegradation, biomimicry, and biomechanical features if it is not bioprintable. The bioprintability of the hydrogel is dependent on several factors, amongst them are processing parameters (such as nozzle gauge, extrusion pressure, printing speed, infill pattern [60], infill density, and infill angle), rheological properties of the hydrogel, printing time, crosslinking mechanism and kinetics, and protection against external stresses [41]. Most importantly, those parameters are also dependent on each other; for instance, a higher printing time is achieved with a higher infill density for each infill pattern and each infill angle. Furthermore, the printing time is reduced when a 90° infill angle is chosen as opposed to a 45° infill angle [92]. All these factors must be carefully considered when depositing bioink in a precise and controlled fashion with the desired spatial and temporal control keeping in mind, the type of bioprinting modality that best fits with the unique properties of each hydrogel. Additionally, the mechanical integrity, stability, and shape fidelity [91] of the 3D-bioprinted tissue-engineered constructs should be maintained for a long period of time to be considered for future clinical translation. 

### 3.4. Biodegradation 

The biodegradation mechanism and kinetics of the biomaterial are key factors when determining whether a bioink is suitable for a specific tissue engineering application or not [86]. Regardless of the choice of tissue engineering application, all bioinks must have degradation rates that are slower than the ability of cells to produce their own extracellular matrix proteins [93], degrade into harmless waste products, and perform the rapid removal of waste products from the body without influencing the viability of encapsulated cells or inducing any physical and chemical changes to the tissue-engineered structures [37]. With regards to biodegradation kinetics, it is known that biomaterials with slow biodegradation properties and good mechanical properties are suitable for hard tissue engineering; however, biomaterials generally used in soft tissue engineering and specifically as sacrificial templating are required to have rapid biodegradation kinetics [43,94]. The degradation of the biomaterials is dependent on several parameters including concentration, temperature, composition, moisture, and many others. Swelling and contractile characteristics are important factors to consider when fabricating tissue-engineered structures since swelling is responsible for the absorption of fluid from the surrounding tissues and contraction is the result of the closure of pores or vessels that are key for the transportation of oxygen and nutrients to the cells. 

### 3.5. Biomechanical and Structural Characteristics 

The selection of biomaterials that make up the 3D-printed tissue constructs should heavily rely on its biomechanical and structural characteristics and whether such characteristics are compatible with the fabrication of the desired tissue type or not (i.e., should closely biomimic the structural and mechanical properties of native tissues). The mechanical stiffness of native neonatal (4.0 and 11.4 kPa) and adult (11.9 to 70 kPa) cardiac tissues is higher compared with native rat embryonic cardiac tissues (cardiac tube < 0.2 kPa) with an increase in mechanical stiffness (50–90 kPa) as a result of ECM remodelling and the proliferation of fibroblasts cells due to the presence of fibrotic tissue present in the infarcted regions of the myocardium [85]. The structural characteristics of the scaffold are heavily dependent on the nature of the scaffold as to whether it is porous or fibrous [95]. The porous nature of the scaffold (high surface/volume ratio) is dependent on the pore size, porosity, and interconnectivity, which are necessary for cellular attachment, migration, and proliferation due to the presence of enough space to perform such cellular functions, in addition to the transport of nutrients and oxygen to the cells. However, this is compromised by the mechanical strength of the scaffold due to the large presence of pores. The ECM-like scaffold biomimics the fibrous nature of the native tissue which is composed of densely packed, aligned fibrillar proteins with nanoscopic topography that could direct cellular behaviour such as the cell attachment, differentiation, and maturation of cardiomyocytes [85,95].

Despite the tissue engineering application, all tissue-engineered scaffolds must possess good mechanical integrity, shape fidelity, functionality, and ability to resist breakdown upon external forces. To achieve the above features, optimization of the following conditions must be achieved: type and concentration of polymer, molecular weight of polymer in the hydrogel, and crosslinking mechanism and kinetics [41]. For instance, a low concentration of polymers is good in terms of cell proliferation and growth; however, 3D-printed tissue constructs will have poor mechanical properties. The structural and biomechanical characteristics must be considered concomitantly with the biodegradation kinetics since they both heavily rely on each other [37]. To enhance the mechanical characteristics of the desired tissue construct, supporting/sacrificial materials could be used during the process of crosslinking to allow enough time for the effective crosslinking to take place or added to the desired tissue constructs for enhanced stability. This acts as a supportive environment until the embedded cells can secrete their own extracellular proteins that are able to withstand the pressure and stresses of the surrounding microenvironment. 

### 3.6. Biocompatibility 

Biomaterials labelled as biocompatible simply mean that they are cell-friendly; do not produce undesirable side effects (for instance, do not degrade into toxic waste products); and provide a 3D microenvironment sufficient for cellular attachment, growth, and proliferation, in addition to taking part in the facilitation of molecular and mechanical signalling pathways and many other functions [37]. In the past, biocompatibility was defined as the safe interaction of non-living materials with living tissues, but not too long ago, this definition was modified to also include the induction of positive cellular responses that allow for healthy and functional bioprinted tissue structures in short- and long-term studies. An example of biocompatible material investigations is the study by Buj-Corral et al. reporting the use of 3D-printed yttria-stabilized zirconia parts encapsulated with human bone marrow-derived mesenchymal stem cells in a hard tissue engineering application [96]. Results showed that high biocompatibility was achieved at 3 mol% yttria-stabilized zirconia, which is a promising result as opposed to using titanium or Cr-Co alloys that can produce a huge health risk due to the release of metal ions into the body. 

### 3.7. Biomimicry 

The biocompatibility of an engineered tissue construct is highly dependent on whether the engineered tissue construct is integrated with biomimetic components like the native tissue’s surroundings or not. If so, this will allow for the endogenous and exogenous cells to enhance their functionality through attachment, migration, and proliferation [41]. Furthermore, the selection of the material, addition of surface ligands to the material, intracellular interactions, design features of the scaffold, and nature of the surrounding 3D microenvironment will greatly influence the morphological changes to the attached cells, therefore affecting the proliferation and differentiation of cells [37]. To fabricate accurate engineered tissue constructs, it is of utmost necessity to comprehend the nature of specific naturally occurring constituents (i.e., extracellular matrix products), their precise location, and their biological function within the tissue of interest. 

All in all, it is imperative to investigate each of the above-mentioned properties when choosing the right bioink for fabricating 3D-bioprinted constructs that exhibit biocompatibility, good rheological property, an appropriate crosslinking mechanism suited towards the application, fast gelation rate, bioprintability, optimum degradation kinetics that are appropriate for the regeneration of defected region of the heart, and that have comparable biomechanical, structural, and compositional properties with the native cardiac environment. 

## 4. Biomaterials 

### 4.1. Natural Biomaterials 

#### 4.1.1. Alginate-Based Hydrogels

Alginate, commonly known as algin or alginic acid, is usually obtained from calcium, magnesium, and sodium alginate salts from the cell walls and intracellular spaces of different brown algae. It is a cheap and soluble biopolymer that is naturally occurring, non-toxic, biocompatible, biodegradable, and a non-immunogenic linear polysaccharide composed of (1–4)-linked β-d-mannuronic (M) and α-l-guluronic acid (G) blocks [97]. The gelation process of alginate is a fast process due to the presence of multivalent cations (i.e., Ca^2+^) and it is speculated that the cations bind to G blocks and M blocks through the generation of ionic interchain bridges [98]. Alginate bioinks have superior rheological properties due to their shear thinning property, which makes alginate an ideal bioink for printing due to its ease of handling and extrusion of continuous filaments whilst keeping the encapsulated cells protected [99]. The molecular weight and concentration of alginate-based hydrogels influences the structural and mechanical properties of the hydrogel; however, those properties can be tuned according to the field of application (e.g., soft or hard tissue engineering) [89]. Alginate is known for its fast ionic crosslinking capacity which makes it one of the most exploited bioinks in any 3D-bioprinted system [99,100]. Alginate-based materials are commonly mixed with other polymers to enhance their structural, mechanical, and biological properties such as gelatin [101], GelMA [102], gelatin-fibrinogen [103], PEGDA [104], PEG [105], RGD modification [63], carbon nanotubes [106], and fibrin [107], among others. In terms of biological function, arginine-glycine-aspartate peptides, heparin-binding peptides, or Matrigel are added to alginate to allow for cardiac regeneration through improved cellular attachment, proliferation, migration, and other cellular functions. Even though unmodified alginate hydrogel has shown great success in supporting the differentiation of human cardiac progenitor cells into mature cardiomyocytes and enhanced gene expression of the early cardiac transcription factors Nkx2.5, Gata-4, and Mef-2c as well as the sarcomeric protein Troponin T [63], there are major drawbacks that are still unmet. These include a limited capacity for cellular proliferation and differentiation, poorly conductive pore walls that hinder cardiac action potential propagation, limited or slow biodegradability, lack of nanofibrous structural features which biomimic the native ECM which is necessary for regulating cellular behaviour, poor electrical conductivity specifically at physiologically relevant frequencies (e.g., 1–5 Hz), and limited mechanical properties which could result in the collapse/rupture of the printed scaffolds due to the stretching–contracting stresses associated with the beating myocardium [70].

In 2012, Duan et al. reported the bioprinting of the first biological aortic heart valve via a multihead microextrusion bioprinter by printing with PEGDA-alginate-based hydrogels consisting of various mechanical properties (stiff hydrogel (E = 74.6 kPa) for the aortic root wall and soft hydrogel (E = 5.3 kPa) for the tri-leaflet valve) [104]. As a result, porcine aortic valve interstitial cells encapsulated in printed heart valves showed high cell viability over a period of 21 days of culture (100% on day 21 compared to 91% on day 1). The following year, the same group reported the printing of an aortic heart valve with alginate-gelatin-based hydrogels seeded with human aortic root sinus smooth muscle cells and porcine aortic valve interstitial cells experiencing high cell viability (>81%) for more than 7 days with no change in mechanical properties of the valve since day 1 of culture [101]. In 2018, Maiullari et al. reported the fabrication of 3D-bioprinted cardiac tissue composed of alginate and PEG-fibrinogen hydrogels encapsulated by human umbilical vein endothelial cells (HUVECs) and induced pluripotent cell-derived cardiomyocytes (iPSC-CMs) using a microfluidic printing head [108]. Results showed that the 3D engineered cardiac tissue was able to form a seamless integration with the host vasculature when placed in vivo. In 2020, Adam Feinberg’s group published the 3D bioprinting of a full-sized adult heart model using the FRESH technique from patient-derived magnetic resonance imaging (MRI) data sets using alginate hydrogel due to the match between the elastic modulus of alginate hydrogel and cardiac tissue [109].

#### 4.1.2. Gelatin-Based Hydrogels

Gelatin-based hydrogels have been widely used due to their appropriate biological properties and tuneable physical characteristics in various biomedical applications. GelMA is a gelatin derivative commonly known as gelatin methacrylate [110], methacrylated gelatin [111], and other names [111]. Gelatin is the end-product of the hydrolysis of collagen (main component of the native ECM in various tissues), therefore it contains many arginine-glycine-aspartic acid (RGD) sequences that are essential for cell attachment and the presence of matrix metalloproteinase-responsive peptide-degradable motifs allows for cell remodelling [112]. GelMA is primarily composed of methacrylamide groups and a few methacrylate groups that are essentially formed through the replacement of the amine groups in gelatin to methacrylic functional groups, which makes it photopolymerizable. This is the reason behind GelMA having superior mechanical properties compared to gelatin. The photopolymerization of GelMA is a result of the formation of covalent crosslinking through the photoinitiation of the radical polymerization of methacryloyl groups. The degree of substitution, GelMA and photoinitiator concentration, and UV exposure time are the major factors that influence the mechanical properties of the GelMA biomaterials [113]. This is supported by its ability to form a gel-like property at room temperature which makes it an excellent biomaterial for 3D printing applications [114]. GelMA hydrogels with versatile characteristics could be synthesized and processed by altering their physical (i.e., elastic modulus, degradation, porosity, and swelling) and cell-responsive properties (survival, proliferation, differentiation, and migration) according to their suitability in the different tissue engineering applications. The resultant GelMA hydrogels are influenced by the direct and inverse proportionality between the above properties [115]. GelMA hydrogels are mixed with many materials including, but not limited to, nanocomposites such as carbon nanotubes [116] and gold nanorods [117] for 3D printing functional cardiac constructs. 

Tomov et al. reported the fabrication of a 3D-bioprinted in vitro model of Tetralogy of Fallot with major aortopulmonary collateral arteries composed of gelatin methacrylate hydrogel, and ultimately, those engineered constructs were perfused and flow was analysed using clinical imaging and computational fluid dynamics [118]. Following that, Cetnar et al. reported the fabrication of 3D-bioprinted, anatomically accurate models of embryonic hearts (day 22) and foetal left ventricles (week 33) composed of gelatin methacrylate hydrogel encapsulated by endothelial cells that were perfused and studied using computational and experimental techniques (ultrasound and magnetic resonance imaging) [119]. Results showed cellular growth and function within the bioprinted constructs across various cardiac geometries and flow.

#### 4.1.3. Collagen-Based Hydrogels

Collagen is one of the most copious proteins available in almost all biological tissues which accounts for approximately ~30% of total protein mass [120]. Collagen belongs to a family of at least 28 proteins with type 1 collagen being the dominant component of the native extracellular matrix in biological tissues. The main function of collagen is to provide mechanical integrity to different parts of the biological tissues and organs [120]. Furthermore, type 1 collagen is an ideal choice for collagen-based hydrogels due to its biocompatibility, abundancy, ease of extraction, biodegradability (enzymatic degradation using matrix metalloproteinases) [121], low immunogenicity, occurrence of natural cell-binding sites, promotion of cell attachment, proliferation, differentiation, migration, and apoptosis [122,123]. However, collagen bioink is known for its slow gelation, low viscosity, and poor mechanical characteristics, which limits its application in tissue engineering despite its superior biocompatibility [124]. There have been several reports of the 3D printing of collagen through in situ crosslinking and high-viscosity collagen bioinks [125,126]. The gel mechanics, gelation kinetics, and printability of collagen bioinks are affected by crosslinking, temperature, and pH [127,128]. By tuning the physical and chemical properties to create fast gelation rates, high viscosity, and enhanced mechanical properties, collagen-based hydrogels could be the ideal bioink in the field of tissue engineering. 

Sanz-Fraile et al. reported the fabrication 3D-bioprinted constructs composed of a collagen–silk hybrid hydrogel encapsulated by bone marrow-derived mesenchymal stem cells for potential use as cardiac patches for myocardial regeneration [129]. Results showed that biocompatibility was maintained due to the presence of a high concentration of collagen and enhanced printability was achieved due to the presence of silk, which yielded bioprinted constructs that exhibited mechanical characteristics like those of natural soft tissues. Kim et al. reported the fabrication of porous 3D cardiac mesh (cMesh) tissue using gelatin-methacryloyl–collagen hydrogel mixed with human cardiomyocytes and human cardiac fibroblasts as a therapeutic method for acute myocardial infarction (AMI) in an animal model [130]. In an in vivo culture, the cMesh tissue showed enhanced long-term graft survival, vessel formation, and stabilization; reduced fibrosis; elevated left ventricle thickness; and improved cardiac function. 

#### 4.1.4. Hyaluronic Acid-Based Hydrogels

Hyaluronic acid (HA), or hyaluronan, is a linear polysaccharide that consists of alternating units of a repeating disaccharide, β-1,4-D-glucuronic acid–β-1,3-N-acetyl-D-glucosamine. HA is a non-sulphated, highly hydrated polyanionic glycosaminoglycan and is most commonly found in different parts of the body, especially the vitreous of the eye and extracellular matrix (ECM) of most tissues (i.e., cartilage, cardiovascular, and neural) [131], which makes it a highly biocompatible biomaterial. HA is commonly combined with other hydrogels such as gelatin and collagen or by chemically crosslinking the pendant reactive groups through addition/condensation chemistry or radical polymerization. This will yield hydrogels with enhanced biological and biomechanical characteristics and long-term stability similar to the ECM of native tissues, which is useful in cardiovascular tissue engineering applications [39]. In 2013, Duan et al. reported the 3D bioprinting of tri-leaflet valve conduits composed of methacrylated hyaluronic acid (Me-HA) and methacrylated gelatin (Me-Gel) encapsulated with human aortic valvular interstitial cells (HAVICs) [132]. This was achieved by optimizing the concentrations and ratios of Me-HA and Me-Gel, which resulted in increased viscosity, lower stiffness, and high cell viability. As a result, the deposition of collagen and glyosaminoglycans was achieved through the bioprinting of anatomically accurate tri-leaflet valve conduits. In 2020, Xu et al. reported the fabrication of 3D-bioprinted small-diameter, heterogenous bilayer blood vessel-like grafts composed of hybrid hydrogels (gelatin methacryloyl, hyaluronic acid, glycerol, and gelatin) at various concentrations and encapsulated with two types of cells across the bilayer of the engineered blood vessel in a single step [133]. The inner tissue layer was composed of a high concentration of gelatin methacryloyl bioink (6% *w*/*v* GelMA, 2% gelatin, 0.3% *w*/*v* HA, and 10% *v*/*v* glycerol) and encapsulated with human umbilical vein endothelial cells (HUVECs), whilst the outer tissue layer was composed of a lower concentration of gelatin methacryloyl bioink (4% *w*/*v* GelMA, 4% gelatin, 0.3% *w*/*v* HA, and 10% *v*/*v* glycerol) and encapsulated with smooth muscle cells (SMCs). Results showed that the bioprinted blood vessel-like grafts biomimicked native blood vessels and exhibited enhanced mechanical strength and maintained high cell viability and proliferation.

#### 4.1.5. Fibrin-Based Hydrogels

Fibrin is a biopolymer composed of several monomers of fibrinogen. Each fibrinogen molecule is made up of two groups of three polypeptide chains (linkage of Aα, Bβ, and γ through six covalent disulfide bonds) [134]. Insoluble fibrin is formed as a result of the cleavage of fibrinogen by thrombin which is later covalently crosslinked using blood coagulation factor XIIIa forming a stable fibrin network that is resistant to enzymatic degradation [135,136]. Fibrin hydrogels have several advantages, including the ability to use the patient’s own blood to produce fibrin for scaffolding, therefore minimizing the risk of immune inflammatory response, having advanced cell adhesion properties [137], and high seeding efficiency. However, the main disadvantages are poor mechanical characteristics and fast degradation before the production of stable printed structures [138]. The fast degradation of fibrin is mostly the result of the secretion of degradation enzymes (plasminogen and matrix metalloproteinases) from the encapsulated cells in the fibrin hydrogel; however, this can be tackled by either the introduction of plasmin inhibitor (aprotinin) to the fibrin hydrogel [139] or the optimization of the pH and the concentrations of fibrinogen and calcium ions (Ca^2^⁺) [140]. The fibrin hydrogels could be combined with other hydrogels to enhance the biomechanical characteristics of the printed tissue-engineered structures. In 2018, Wang et al. reported the 3D bioprinting of cardiac tissue constructs from cardiomyocyte-laden fibrin hydrogels used as the primary hydrogel with a sacrificial hydrogel (composed of gelatin and HA) and support material (poly(Ɛ-caprolactone)) [141]. The functional bioprinted cardiac constructs showed spontaneous synchronous contraction from dense and organized cardiac muscle bundles in an in vitro culture, confirming cardiac tissue development and maturation. Freeman et al. reported the fabrication of 3D-bioprinted vascular conduits composed of a gelatin-fibrin hydrogel using a novel rotary 3D bioprinter [142]. The presence of heat-treated gelatin in the hydrogel mixture converted the unprintable fibrinogen into printable hydrogel due to gelatin’s rheological properties, especially in terms of its shear-thinning property. After 2 months of cell culture, collagen deposition and mechanical properties were enhanced in terms of both circumferential and axial elastic moduli in addition to the burst pressure of the engineered vessel reaching 1110 mmHg. 

#### 4.1.6. Matrigel-Based Hydrogels

Matrigel is an ECM-based hydrogel containing the important growth factors and peptides needed for cell growth, adhesion, and proliferation in 3D constructs [143]. Nevertheless, Matrigel is an expensive material [41] and has poor mechanical characteristics [144], which makes it a non-ideal stand-alone bioink for 3D bioprinting. In 2015, Hinton et al. reported the 3D printing of cell-laden hydrogel composed of fibrinogen–collagen–Matrigel using the freeform reversible embedding of suspended hydrogels technique to fabricate cardiac constructs with high cell viability (99.7%) [81]. In 2015, Ribeiro et al. reported that single cardiomyocytes (presented as a highly organized cellular architecture) experienced enhanced alignment, synchronous contraction, and maturation when cultured in rectangular Matrigel micropatterns [145]. In another report, 3D bioprinting of tissue constructs using cell-laden hybrid Matrigel—agarose hydrogel showed super performance compared with stand-alone Matrigel hydrogel since agarose was able to maintain the stability of the 3D-printed structures, whereas Matrigel provided an appropriate environment for cell growth and adhesion [146].

#### 4.1.7. Agarose-Based Hydrogels

Agarose is a naturally derived polysaccharide composed of reiterating monomeric units of D-galactose and 3,6-anhydro-L-galactopyranose, thereby creating a densely packed gel with low porosity through the thermal process of chain entanglements [121]. It is a bio-inert and biocompatible material that has been extensively used as a mould material when using different fabrication methods such as 3D printing and casting techniques [147]. The main advantage of using agarose hydrogels is their distinguished mechanical characteristics which allow for 3D-printed constructs to maintain their shape fidelity until the encapsulated cells grow, proliferate, mature, and assemble into the desired tissues [146,147]. The main disadvantages are the non-biodegradability and requirement of low temperature for gelation or the use of heated dispenser units. In 2012, Daniela et al. reported the use of hydrophobic high-density fluorocarbon as a mechanically supportive template for the fabrication of 3D-bioprinted constructs of human mesenchymal stem cell-laden agarose hydrogel [148]. As such, the encapsulated cells remained viable for 21 days with significant cell proliferation, production of their own extracellular matrix, and finally, the 3D-printed constructs remained stable for more than six months. Bertassoni et al. reported the fabrication of perfusable, functional microchannel networks within cell-encapsulated photo-crosslinkable hydrogels such as methacrylated gelatin (GelMA), star poly(ethylene glycol-co-lactide) acrylate (SPELA), poly(ethylene glycol) dimethacrylate (PEGDMA), and poly(ethylene glycol) diacrylate (PEGDA) using the 3D micro-moulding technique whilst using a 3D extrusion printer to print agarose fibres as a sacrificial template [121]. It has been demonstrated that there was 90% cellular viability and differentiation; furthermore, endothelial monolayers were successfully formed within the tissue-engineered structure. This was also achieved by printing agarose microchannel networks (thickness of the wall was 200 µm), thereby allowing for an adequate supply of nutrients and oxygen to the tissue-engineered structure.

#### 4.1.8. Chitosan-Based Hydrogels

Chitosan is a linear polysaccharide molecule composed of random β-(1-4)-linked D-glucosamine and N-acetyl-D-glucosamine units and it is obtained from deacetylation of chitin (second most abundant carbohydrate polymer) [41,149,150]. It has antibacterial properties and could be used as a bioactive delivery vehicle [151]. The main advantages of chitosan hydrogels are their biocompatibility, biodegradability, intrinsic cellular interactions, and low cost of sourcing [150,152], which makes them an appropriate matrix for 3D structures [149]. However, the main drawback is the unstable biomechanical characteristics which limit their use in the 3D bioprinting of scaled-up structures [41]. In 2013, Zhang et al. reported the fabrication of 3D-bioprinted perfusable vessel-like microfluidic channels composed of chitosan hydrogels using a co-axial nozzle system where, briefly, the crosslinking solution and chitosan solutions were simultaneously printed allowing for the rapid gelation of chitosan, thereby producing hollow tubular structures [153].

#### 4.1.9. Decellularized Extracellular Matrix (dECM)-Based Hydrogels

The Decellularized Extra Cellular Matrix (dECM) is formed through the removal of cells leaving behind the extracellular matrix components. The extracellular matrix components are secreted by cells, thereby capturing the native tissue environment which provides support for cell attachment, growth, proliferation, signalling, and tissue development. This is conducted through the presence of growth factors, matricellular proteins, and complex ultrastructural compositions [154]. Even after the remodelling of dECM scaffolds by the host tissue, the degradation products have been shown to influence endogenous cell activity which might play a major role in tackling the issue of the limited regenerative capacity of the heart. More evidence proves that the plasticity and regenerative capacity of endogenous cells and the anatomic structure of the heart are the main reasons behind the induction of tissue remodelling and improvement in the function due to dECM scaffolds [155]. dECM is prepared by using chemical, physical, and enzymatic methods without interfering with the extracellular matrix components (98% removal of cellular components is considered successful) and then solubilized by being converted into a gel-like form that is suitable for use as a bioink [41]. The main advantages are the retainment of the mechanical properties of the extracellular matrix, which provide an ideal 3D microenvironment for cells to proliferate and differentiate, and the ability to decellularize the extracellular matrix from the same tissue of interest to use as a bioink, thereby enhancing the interaction between the printed structures and the target tissue [37]. Furthermore, the dECM scaffolds could be used as a stand-alone therapy or as a vehicle used to deliver therapeutic agents to the heart which can specifically help in tackling the issue related to cell survival and engraftment because of the hostile nature of the post-MI microenvironment [154,156]. All in all, this helps in the pro-regenerative signalling, which ultimately leads to cardiac tissue repair and remodelling. However, the major drawback is the scarce availability of native tissues needed to produce small volumes of bioink, thereby increasing its cost and reducing its potential for scalability [41]. In addition, the encapsulated cells in the dECM hydrogel release MMPS which enzymatically degrades the dECM [41]. In 3D bioprinting, the major drawback for the use of dECM is the weak mechanical properties, which make it non-ideal for use as a primary bioink but rather a secondary bioink. Jang et al. reported the successful fabrication of a 3D-printed pre-vascularized cardiac progenitor cell-laden dECM cardiac patch which showed enhanced vascularization and tissue remodelling and reduced cardiac hypertrophy and fibrosis at the myocardial infarcted sites [157]. Basara et al. reported the fabrication of tissue-like beating 3D-bioprinted constructs composed of decellularized human heart ECM (dhECM) with either gelatin methacryloyl (GelMA) or GelMA-methacrylated hyaluronic acid (MeHA) hydrogels mixed with human induced pluripotent stem cell-derived cardiomyocytes (iCMs) and human cardiac fibroblasts (hCFs) [158]. The functional, 3D-bioprinted constructs exhibited striated sarcomeric alpha-actinin and connexin 43 expression and, as such, as a proof of concept, a myocardial infarct boundary region was created by controlling the mechanical properties along with the cellular and macromolecular content. Das et al. reported the fabrication of a 3D-bioprinted cardiac tissue model composed of a heart tissue-derived extracellular matrix (hdECM) hydrogel and collagen hydrogels mixed with primary cardiomyocytes, where the enhanced maturation of cardiomyocytes in hdECM as opposed to the collagen group was shown [159]. Furthermore, the early differentiation of cardiomyocytes was seen when a low concentration of hdECM bioink was used, further proving that the matrix and the culture microenvironment can be limiting factors for cell–material interactions that affect cardiomyocyte maturation.

#### 4.1.10. Silk Fibroin-Based Hydrogels

Silk is known to be a strong and tough material originally obtained from silkworms and spiders (native and recombinant silk proteins) [160]. Silk is used in 3D printing by easily controlling the shear force (pneumatic or piston) which will induce a sol-to-gel transition, thereby producing 3D-printed structures exhibiting super mechanical characteristics and slow degradation in vivo. Silkworm silk is the most frequently researched type of silk due to its ease of availability. It is composed of heavy (≈350 kDa) and light (≈26 kDa) amino acid chains with a glycoprotein (P25) in which heavy and light chains are linked by a strong disulfide bond, whilst non-covalent interactions are present between P25. Spider silks have superior mechanical properties compared with silkworm silk due to the sequencing of highly repetitive amino acids forming amphipathic, stable protein structures. The main advantages of silk fibroin are its ability to blend with other biomaterials, enhanced cell adhesion factors, ease of structural modification, controlled degradation, and the ease of inducing the sol-to-gel transition without the need for chemical or physical crosslinking. The main disadvantages are differences in batch-to-batch products due to the wide variations in the amino acid sequences amongst different species and optimization of the rheological (i.e., viscosity and surface tension) properties of the bioink [160]. In 2015, Das et al. reported the 3D bioprinting of clinically relevant-sized tissue constructs composed of human nasal inferior turbinate tissue-derived mesenchymal progenitor cell-laden silk fibroin–gelatin hydrogels [161]. The multilineage differentiation (chondrogenic, osteogenic, and adipogenic), ECM production of the encapsulated stem cells, and tissue formation of choice were successfully achieved. In another study, Rodriguez et al. reported the 3D printing of silk, gelatin, and glycerol bioinks to increase the printing resolution to meet patients’ specific needs in the field of soft tissue regeneration [162]. The composite bioinks have been assessed for their stability, biocompatibility, and cellular and tissue integration in vitro and in vivo through the optimization of the concentration and ratio between individual bioinks. Kim et al. reported the fabrication of 3D-bioprinted structures (heart, vessel, brain, trachea, and ear) composed of silk fibroin-methacrylate (Sil-MA) using the digital light processing 3D bioprinting technique [163]. This approach showed good results in terms of enhanced mechanical and rheological characteristics due to the varying concentrations of Sil-MA, elevated stability, and desirable biocompatibility of the engineered structures.

### 4.2. Synthetic Biomaterials

#### 4.2.1. Nanocomposite-Based Biomaterials

The application of nanotechnology in biomedical applications has recently attracted the attention of many researchers due to several factors including, but not limited to, the antimicrobial activity of nanomaterials, thereby it has been used in drug delivery, the repair of damaged tissues [74], bioactuators, and biosensors [164]. The common types of nanocomposite biomaterials are ceramic matrix, metal matrix, and polymer matrix, specifically the latter, which can exhibit porous, covalently, or physically crosslinked bondages with incorporated nanomaterials, thereby exhibiting various tunability properties, good thermal conductivity, enhanced mechanical strength, and good biocompatibility and biodegradability characteristics, which makes it superior compared with other nanocomposites [164,165]. The incorporated nanomaterials within the porous polymeric matrices are composed of a wide range including, but not limited to, carbon-based nanomaterials (such as carbon nanotubes (CNTs), graphite oxide, and graphene), polymeric nanomaterials (such as polymeric nanoparticles and hyperbranched polyesters), and metal/metal-oxide nanostructures (gold, silver, and iron oxide) [164,166,167]. The target of such a combination is to produce nanocomposites that are electrically conductive, mechanically robust, and biologically functional with tuneable behaviour that befits various types of complex tissue engineering applications. Such a combination is an asset to the development of tissue fabrication, neovascularization, and the treatment of diseases [74]. Specifically, the incorporation of nanomaterials to polymeric bioinks arose from the incapability of polymeric biomaterials to exhibit conductivity behaviour (i.e., electrical coupling) between adjacent cardiac cells in 3D-bioprinted cardiac tissues. Furthermore, the presence of nanomaterials can enhance the shear-thinning properties and tweak the stiffness, thereby improving the degradation rate of the polymeric structures [165]. As such, Zhu et al. reported the use of 3D-bioprinted gold nanorods (GNRs) incorporated into gelatin methacryloyl (GelMA) hydrogel to create 3D functional cardiac tissue constructs [117]. The incorporation of GNRs allowed for enhanced electrical propagation between cardiac cells (producing synchronous beating behaviour within cardiac constructs), thereby enhancing the proliferation of cells, and GelMA has integrin-binding motifs which allowed for cellular interactions/attachment. Li et al. reported the fabrication of 3D-bioprinted hollow blood vessels composed of a gelatin, sodium alginate, and carbon nanotube hybrid hydrogel encapsulated by mouse epidermal fibroblasts (inner and outer walls of the tubular scaffold) [168]. Results showed an increase in mechanical properties of the scaffold, enhanced surface smoothness, and signs of reduced cytotoxicity which fulfils the biomimetic vasculature criteria. Afghah et al. reported the fabrication of a 3D-bioprinted complex, large-scale vascular scaffold composed of NIH-3TS cell-laden alginate-based hydrogel printed on a support bath consisting of Pluronic F127-nanoclay in addition to calcium ions, used to improve the viscoelastic properties of the polymer–nanoclay composite and a crosslinker agent for alginate hydrogel [169]. Results showed an enhancement in viscoelastic properties, structural integrity, interconnectivity between the channels, and high cellular viability of NIH-3TS cells (above 80%) within the hydrogel.

#### 4.2.2. Polyethylene Glycol (PEG) Derivative-Based Hydrogels

PEG-based hydrogels have been extensively used in the medical and nonpharmaceutical fields [41]. They are linear polyether hydrophilic materials that can be combined with various biomolecules including, but not limited to, proteins, enzymes, and liposomes [41]. They are known for being biocompatible, non-immunogenic, and non-toxic biomaterials [93]. The mechanical properties of PEG can be altered through ionic crosslinking; however, PEG exhibits poor mechanical strengths compared to methacrylated or diacrylated PEG produced through the photopolymerization of PEG [41]. PEG-based hydrogels have shear thinning properties which makes them a suitable candidate for sacrificial templating and can occasionally allow for cellular activity (only for certain cell types such as osteoblasts and chondrocytes without the need for additional supportive biomolecules [41]) due to their significant mechanical properties [170]. The main drawbacks are their limited or slow biodegradability [61] and protein adsorption repellent properties [41]. Additional modifications to PEG-derivative hydrogels include RGD peptides [171] or mixing with other hydrogels such as alginate [104,105,121,172], gelatin derivatives [173], and hyaluronic acid derivatives [174] in order to enhance cellular attachment, migration, and proliferation. In 2010, Skardal et al. reported the 3D bioprinting of bioartificial vessel-like grafts composed of tetrahedral PEG derivatives crosslinked with hyaluronic acid derivatives and thiolated gelatin [173]. The resultant extrudable hydrogels exhibited high shear storage moduli and enhanced cellular growth and proliferation compared with PEGDA crosslinked with synthetic extracellular matrices of similar composition. Miller et al. reported the fabrication of a perfusable, vascular channel with a diameter of 150 µm encapsulated with human umbilical vein endothelial cells (HUVECs) using a thermal-based 3D bioprinter [175]. This 3D engineered structure was formed using a sacrificial lattice (carbohydrate glass: sucrose, glucose, and dextran) surrounded by a hydrogel consisting of fibrin, alginate, agarose, Matrigel, or PEG-based biomaterial. The encapsulated cells survived, spread, and migrated, thereby reaching confluence within one day of seeding. 

#### 4.2.3. Pluronic F127-Based Hydrogels

Pluronic F127 is a synthetic polymer poloxamer based on a non-ionic triblock copolymer that is composed of two hydrophilic blocks and a central hydrophobic block, thereby making it an effective surfactant [176]. It exhibits shear thinning behaviour and has a thermosensitive nature, which makes it a suitable candidate for sacrificial templating [80,177], especially in the fabrication of complex perfusable channels within cell-laden constructs. Pluronic-based hydrogels can have superior mechanical strengths and structural integrity when they are physically, chemically, and enzymatically crosslinked or combined with other hydrogels [41,178]. They are very useful in extrusion-based bioprinting due to their significant gelation at room temperature; however, it has not been used in inkjet-based bioprinting and laser-assisted-based bioprinting due to its high viscosity and inability to transfer to thermal energy and kinetic energy, respectively [41]. In 2014, Kolesky et al. reported the 3D bioprinting of perfusable and vascularized networks fabricated from sacrificial Pluronic F127 inks and cell-laden methacrylated gelatin hydrogels [177]. Using the differences in thermo-reversible gelation between the bioinks, sacrificial Pluronic F127 was removed at low temperatures, creating hollow bifurcated microchannels which were endothelialized with human umbilical vein endothelial cells (HUVECs) showcasing a cell viability of greater than 95% after 2 days of cell culture. Suntornnond et al. reported the 3D bioprinting of a perfusable, complex vasculature network composed of a hybrid hydrogel (Pluronic 127 and GelMA) and Pluronic 127 as the support hydrogel, encapsulated with L929 fibroblasts and human vein endothelial cells (HUVECs) in a single-step bioprinting process [179]. Results showed that the new hydrogel composite allowed for L929 cells to attach to the engineered structure and proliferate, in addition to supporting HUVECs in releasing their extracellular matrices over a 7-day period which might have led to angiogenesis and vascular branching. All in all, the hydrogel composite showed superior characteristics in terms of printability, shape integrity of the engineered structures and biocompatibility.

The advantages and disadvantages of natural and synthetic biomaterials are summarized below in Table 1. 

## 5. Cell Sources and Biomolecules

There is a broad spectrum of cells used in cardiac tissue engineering; however, the appropriate selection of cell types should be based on the achievement of a balanced cardiomyogenesis and vasculogenesis through the fabrication of viable and functional tissue constructs. Generally, native tissues are composed of various cell types, each with a specific biological function, therefore, it is important to consider this when fabricating tissue-engineered constructs that biomimic the native tissues. Tissues are composed of functional cell types and cells that provide support, structural, or barrier abilities (necessary for vascularization) or provide stem cell maintenance (self-renewal, growth, migration, proliferation, etc.) and differentiation [37]. 

The heart comprises several cell types including cardiomyocytes (CMs: 25–30% of the heart cells) and the rest (70% of the heart cells) are blood and lymphatic endothelial cells (ECs), cardiac fibroblasts (CFs), vascular smooth muscle cells (VSMCs), cardiac progenitor cells (CPCs), pericytes, and immune cells. CMs are the contractile elements which carry blood, oxygen, and nutrients. ECs are responsible for regulating the vasomotor tone and monitoring angiogenesis, and they are metabolically active. CFs are responsible for sustaining the extracellular matrix and balancing the degradation–deposition process. VSMCs and pericytes are responsible for controlling the blood flow in cardiac vascular networks. CPCs are related to the early stage of cardiac mesoderm and have the ability to differentiate in vitro and in vivo into three cell layers, ECs, VSMCs, and CMs, due to its multipotency [180]. 

CMs are the primary cell type responsible for contraction in the myocardium. CMs are necessary for optimal cardiac pump function throughout a person’s lifetime; however, they do not have the capability to self-regenerate and tend to decrease with advancement in age. Adult CMs tend to keep their rod-shape structure after isolation; however, they are unpredictable in extended-term cell culture with a restricted proliferation rate after isolation [181]. As such, CMs from neonatal and foetal sources could be recruited with good cellular proliferation and functional beating behaviour; however, they have various restrictions including immunogenicity due to their allogenic source, partial ability to expand ex vivo, malignancy, restricted accessibility and cellular viability in hypoxic conditions, and ethical concerns [93]. 

ECs as a primary source are either isolated from the human umbilical vein vascular wall (through collagenase treatment) or other human vessel biopsies or differentiated from induced pluripotent stem cells (iPSCs). They make up the endothelium lining of the vascular network [21] as well as coronary blood vessels and act as a barrier between the blood and vascular walls. ECs also take part in regulating cardiac contractility and remodelling through the induction of hypertrophy in CMs. When cocultured with VSMCs, collectively, they can support clinically relevant tissue-engineered scaffolds with oxygen and nutrient exchange [181]. 

CFs as a primary source are isolated from myocardial tissue or differentiated from iPSCs [181]. In addition to the secretion of the extracellular matrix through crosstalk between CFs and CMs, thereby providing structural support to the myocardium, they also influence the biochemical and electrical characteristics of the myocardium [182]. Furthermore, the maturation of the CFs has a significant effect on the functionality of CM. 

VSMCs as a primary source are extracted from the medial layer of the blood vessels containing endothelial cells and VSMCs or are differentiated from iPSCs. In addition to their main role as regulators of blood pressure and flow, they also take part in the vascular development and release ECM proteins needed for the remodelling of the vascular walls upon myocardial damage [181,183]. 

For stem cells, four main categories of cells based on their source of origin will be discussed: iPSCs, embryonic stem cells (ESCs), adult stem cells, and perinatal stem cells [13,15]. Stem cells have good expansion capacity in an undifferentiated state, the ability to differentiate into various functional tissue-specific cell phenotypes, and the ability to self-renew [37,184]. 

iPSCs are cell lines that have been obtained from the ex vivo reprogramming of adult somatic cells and they have the pluripotent capacity to differentiate into cells of the three primary embryonal germ layers (almost all cells of the body) [13,185]. They are reprogrammed by four transcription factors (Oct3/4, Sox2, c-Myc, and Klf4) and fibroblasts are the most common type of somatic cell used for the generation of iPSCs [184]. They have high plasticity and ability to maintain their undifferentiated state in vitro over a high number of passages (i.e., 80 passages) before transplantation [37]. From an ethical perspective, it is safer to use iPSCs than ESCs especially in clinical settings where autologous transplantation is preferred. However, their clinical use is limited by the probable risk of tumorigenesis due to the integration of viral vectors into the genomes [184,186]. 

ESCs originate from the inner cell mass of a blastocyst and have a high expansion capacity and pluripotency, which makes them theoretically able, like iPSCs, to differentiate into cells of the three primary embryonal germ layers [14,185]. However, their clinical use is debatable due to ethical concerns related to the critical isolation process which damages the fertilized human embryo [15] and probable high risk of teratoma formation (due to undifferentiated cells that form tumours) [14,184,186]. 

Adult stem cells are the preferred type of cells to use in preclinical and clinical settings as opposed to iPSCs and ESC, due to both their safety reasons and ease of accessibility from adult tissues. Various types of stem cells such as ECs and endothelial progenitor cells (EPCs), mesenchymal stem cells (MSCs), cardiac stem cells and others are derived from post-natal tissues [13]. Whilst ECs are derived from peripheral veins such as saphenous or forearm veins, EPCs are derived from the purification of peripheral blood or bone marrow [184]. EPCs have the potential to proliferate and generate endothelial cells in in vitro and in vivo settings especially in the damaged heart regions (i.e., ischemic myocardium) through vasculogenesis and endothelialization of the tissue-engineered grafts. MSCs are stem cells derived from multiple adult tissues such as skeletal muscle, bone marrow, heart, and adipose tissues. They have the potential to be used in clinical settings due to their multipotency; limited differentiation in vitro into cells of mesodermal lineage such as myocytes, osteoblasts, chondroblasts, fibroblasts, adipocytes, and other stromal entities [185,187]; great expansion capacity; ease of accessibility; good angiogenic properties; and immunoregulatory activity. They could be extracted from autologous sources such as infants/children suffering from CHD and used in their treatment via stem cell therapy or tissue engineering, thereby reducing the risk of immunogenicity. When MSCs are implanted in vivo, they can differentiate into cardiomyocytes and vascular cells and they can stimulate the regeneration of endogenous cardiomyocytes and vascular cells through paracrine mechanisms [13]. MSCs have immunomodulatory, anti-inflammatory, and trophic properties through the release of soluble factors and direct cell-to-cell interaction with immune cells [185], which makes them attractive in clinical applications. However, MSCs are isolated through invasive procedures where a large amount of tissues is used to isolate an adequate number of cells due to their limited expansion potential compared to ESCs [15]. Cardiac stem cells (CSCs) pose as an alternative to MSCs, and they are obtained from the stroma of different tissues. Less than 20 years ago, it was shown that the generation of new, mature cardiovascular cells in the heart originated from undifferentiated cells (i.e., cardiac stem cells present since birth), especially during myocardial infarction [13,188]. CSCs have the capacity to be self-renewed, clonogenic, and multipotent through the differentiation to all types of cardiovascular cell types [13,184]. In 2010, Mishra et al. discovered that the resident cardiac progenitor stem cells are most abundant and have the capacity to proliferate during the neonatal period; however, they undergo a rapid declination with age [189]. It was also suggested that the cardiac progenitor stem cells provide an enhanced regenerative capacity in neonates due to their expression of high levels of Stem Cell Factor-receptor (SCFR or c-Kit); however, whether the cardiomyogenic differentiation by cardiac progenitor stem cells is a result of directed regeneration and/or paracrine effects is still unknown. 

Perinatal stem cells are the stem cells isolated from perinatal tissues such as umbilical cord blood (UCB), the umbilical cord (this includes blood vessels and Wharton’s jelly matrix), and the placenta from the 20th week of pregnancy up to 4th week of life [15]. Those tissues are generally discarded as medical waste after childbirth and, as such, they serve as an ideal source of cells due to the ease and practicality of the non-invasive approach for tissue collection. UCB presents a good source of hematopoietic stem cells and MSCs, where MSCs are isolated from the UCB through adherence to the plastic when cultured in specific media. MSCs are also isolated from the Wharton’s jelly in the umbilical cord which is composed of a gelatinous substance (mainly made up of hyaluronic acid and chondroitin sulphate) [15]. Placental cells are also derived from the placenta, an organ situated between the foetus and the mother where gas exchange, waste disposal, and nutrient uptake take place. Those cells are categorized into cell populations such as mesenchymal cells and progenitor cells and they could be used as a good source of cells in paediatric tissue engineering applications. The placental cells exhibit paracrine properties such as anti-fibrotic, antibacterial, and anti-inflammatory properties [190]; however, they cannot be used in patients who were diagnosed with CHD at a later stage in their lives [184]. 

Biomolecules such as growth factors (GFs), extracellular vesicles (EVs), and microRNAs have been greatly studied due to their significant contributions in achieving direct therapeutic effects in the heart via paracrine mechanisms. GFs such as vascular endothelial growth factor (VEGF), hepatocyte growth factor (HGF), and insulin-like growth factor-1 (IGF-1) have been shown to modulate the growth, survival, and migration of CMs if accurately delivered to the defected/damaged region in the heart [191]. EVs are minute, round-shaped vesicles with a diameter between 30 and 150 nm whose main function is to modulate intercellular mechanisms including cellular communication. They are categorized into three types: microvesicles, apoptotic bodies, and exosomes. Amongst all the paracrine effectors, exosomes released from stem cells have been intensively researched in pre-clinical settings and the results prove that the exosomes rather than the stem cells alone are the sole reason behind the treatment of the defects/damage in the heart. However, due to the short half-life of the exosomes, the long-term effects of using such technology are unknown. MicroRNAs are another paracrine effector known for their role in modulating downstream pathways of messenger RNA via RNA interference; specifically, the fate of cardiomyocytes is influenced by microRNA networks such as miR-24, miR-199a, and miR-590. The single-stranded RNA composed of 20–25 nucleotides is unstable when present in an in vivo environment, thereby posing great challenges as to how to deliver and keep it in the targeted site. Suggested methods including chemical modification and carrier development have been studied and the results are promising; however, more research must be conducted to further understand the impact of such technology on the treatment of defected/damaged regions in the heart [191].

In general, since there are many cell types each with a specific biological function, it is important when creating 3D-bioprinted constructs to choose the appropriate cell types that resemble the native cardiac environment. This is coupled with biomolecules to ultimately achieve a maximum therapeutic effect in the heart. An illustration of different components that could take part in the fabrication of 3D-bioprinted structures is shown below in Figure 3. Those components include wide selections of biomaterials, cell sources, and biomolecules. The selection of each component is dependent on the application with an aim to closely biomimic the native cardiac environment. 

## 6. Vascularization and Angiogenesis

Vasculogenesis and angiogenesis are two distinct mechanisms that take place in human embryos and adult tissues in which new vascular networks are formed. Vasculogenesis is based on the de novo generation of new blood vessels from either vascular progenitor cells or capillary plexus formation due to circulating EPCs. Subsequently, angiogenesis is followed by vasculogenesis where new vessels are formed and extended in various directions because of pre-existing blood vessels as shown in Figure 4. The various ways of forming vascularization within tissue constructs are the fabrication of pre-vascularized constructs and implanting them within the defected/damaged site, gaining support from the host microvascular networks to allow for in-growth of the blood vessels within the tissue engineered constructs or allowing for the formation of a new blood circulation system at the defected/damaged site through stem cell therapy. Additionally, there are diverse ways of promoting angiogenesis either by the formation of rapid perfusion within the implanted construct through the development of interconnections between the pre-vascularized construct and host microcirculatory network or allowing for in-growth of the new blood vessels within the tissue-engineered constructs with the support of host network [192].

The field of vascularization in tissue engineering is challenging due to the inability to maintain metabolic processes and gaseous exchange within thick, functional tissue-engineered structures due to poor incorporation of the tissue-engineered structure with the host vasculature and/or the absence of nutrient pathways/channels within the engineered structures to provide access to nutrients and oxygen to support tissue-specific functions [193]. To successfully carry out stable, long-term metabolic processes and gaseous exchange within the tissue-engineered construct, a limited diffusion pathway of 100–200 µm must be guaranteed, or else the cell viability will decrease and, as a result, the tissue-engineered construct will fail to function. To counter this problem, porous tissue scaffolds could be designed in a manner which allows for enhanced nutrient uptake and gaseous exchange within the tissue-engineered constructs and the introduction of oxygen delivery processes to cells in an efficient fashion. Three-dimensional printing can be used as a biofabrication tool to print vascularized tissues either through the fabrication of perfusable vessels that can be surgically connected with the host blood vessel and/or in vitro seeding with vascular cell types for angiogenic sprouting [192,193,194]. To 3D bioprint functional vascularized tissues, the printing resolution must be within or less than 100–200 µm to maintain high cell viability and support the formation of new blood vessels [121]. There are two methods of 3D bioprinting: the first method known as indirect printing, where the scaffold is deposited first then seeded with cells later, and the second method involves the concomitant direct printing of the tissue structure with cell-encapsulated materials [192], [67]. Both methods have the same target: the ability to print porous structures with high resolution to ensure the supply of nutrients and oxygen to the least-accessible areas in the tissue structure (i.e., centre of the printed structure) [192].

Yeung et al. reported the fabrication of a customized, branched vascular graft using 3D printing and electrospinning techniques for pulmonary artery reconstruction in a porcine model [195]. Computational flow dynamics and magnetic resonance imaging assisted in the design of the customized vascular graft. Results showed that the explanted graft had similar hemodynamic performance in terms of wall shear stress and pressure drop from the main pulmonary artery to the branch pulmonary artery and to the native vessel, and the formation of the endothelial layer and smooth muscle layer on the graft had no signs of calcification. Fukunishi et al. reported the fabrication of customized, acellular tissue-engineered vascular grafts using 3D printing and electrospinning techniques, and they were used as inferior vena cava interposition conduits in a sheep model [196]. Results showed that the explanted graft exhibited patency without any signs of aneurysm or ectopic calcification after 6 months of implantation. Furthermore, there was a formation of an endothelial cell layer and smooth muscle cell layer and the deposition of an extracellular matrix. There was also a significant decrease in pressure gradients over 3–6 months with comparable mechanical characteristics as the native inferior vena cava. As previously mentioned, Miller et al. reported the 3D printing of tough 3D filament networks of carbohydrate glass composed of dextran incorporated with a sucrose–glucose mixture to form cylindrical structures that were later seeded with endothelial cells, thereby creating functional perfusable, vascularized networks [175]. Before endothelization, the carbohydrate glass lattice skeleton was encased in cell-loaded hydrogel matrices composed of fibrin, Matrigel, and alginate and then the skeleton was dissolved away in cell media, leaving behind 150–750 µm diameter-sized interconnected microchannel networks. It was also shown that the perfusable vascular channels within the tissue-engineered structures enabled the metabolic function of primary rat hepatocytes, thereby inhibiting necrosis in the core. Kolesky et al. reported the fabrication of large-sized perfusable vascularized networks (centimetre-sized) through the use of Pluronic F127 mixed with thrombin as a sacrificial hydrogel to create hollow microchannels that were initially encapsulated with human neonatal dermal fibroblast and mesenchymal stem cell-loaded gelatin-fibrinogen hydrogel [197]. Before the dissolution of Pluronic F127, thrombin was used to rapidly gel the surrounding cell-encapsulated fibrinogen hydrogel; furthermore, the printed structures were additionally encased by cell-free gelatin-fibrinogen hydrogel composed of thrombin and transglutaminase used for secondary crosslinking of the remaining gelatin-fibrinogen hydrogel. After the dissolution of the Pluronic F127 hydrogel, the seeded HUVEC channels allowed for the mesenchymal stem cells to differentiate into osteocytes for 6 weeks.

Inkjet printing has also been used to fabricate vascularized tissue constructs. Nakamura et al. reported the inkjet printing of intact, stable fibres and tubular structures composed of alginate- and fibrin-based hydrogels that were initially dropped into a calcium chloride bath solution in order to ionically crosslink the alginate solution [198]. Such a technique is superior compared with extrusion-based printing in terms of printing resolution, where tubular structures with wall thicknesses ranging from 35 to 40 µm and inner diameters ranging from 30 to 200 µm were successfully fabricated. Nevertheless, Guerra et al. reported the fabrication of extrusion-based 3D-printed vascular stents composed of composite biomaterial (polylactide acid (PLA)/polycaprolactone (PCL)) and encapsulated with 3T3 cells [199]. Results showed that the extrusion-based technique (using Fused Deposition Modelling) was able to 3D print PCL/PLA vascular stents with 85–95% accuracy with good biocompatibility and ability to degrade and succeed when undergoing dynamic mechanical and expansion tests, hence paving the way towards their use as biodegradable stents. Laser printing has been used to fabricate vascularized tissue-engineered constructs; however, it has not been extensively used compared with other printing techniques. Xiong et al. reported the freeform bioprinting of straight and Y-shaped overhung structures using an 8% alginate solution and 2% alginate-based mouse fibroblast suspension with an achievement of above 60% cell viability after 1 day of incubation [200]. 

Overall, regardless of the method of formation of the vascularization (pre-vascularization of the 3D-bioprinted constructs prior to implantation in the defected region of the heart or post-vascularization triggered by host cardiac environment), thickness of the vascularized tissue, or method of 3D bioprinting the vascularized tissue, it is important to enable a pathway for cells to receive/exchange nutrients, oxygen, and waste products to ultimately help restore function in the damaged cardiac tissue. 

## 7. Four-Dimensional Printing

Four-dimensional printing is an emerging field within various disciplines such tissue engineering, regenerative medicine, biomedical devices, material science, chemistry, basic sciences, and many other applications where time as a fourth dimension [32] (three dimensions include the *x*, *y*, and *z* axes) is included in the 3D printing fabrication process. In tissue engineering, there are several shortcomings with regards to 3D-printed tissue constructs since it is not guaranteed that the implanted constructs will work and grow like a native tissue [201]. As such, the first type of 4D printing techniques tackles these challenges through the implementation of suitable smart materials that undergo structural and functional changes as a result of internal/external stimuli (i.e., temperature, pH, moisture, magnetic field, etc.) when present in in vitro and in vivo conditions, thus overcoming the static nature of 3D bioprinting [202]. Such changes are manifested in the folding of engineered tissue onto a substrate (i.e., responsive hydrogels) to acquire its desired shape. Soft active materials belong to a broad class of smart materials that are characterized by the capability of experiencing enhanced elastic deformation in response to external stimuli such as heat, light, and electricity. These materials encompass shape memory polymers, hydrogels, and dielectric elastomers [203]. 

This technology is nascent in the medical field. Therefore, there is room for creating and innovating new methods or the refinement of old methods to challenge the complex medical problems. This is conducted through the implementation of advanced material-based technology capable of fabricating biomedical implants and devices composed of materials with superior performance, efficiency, and quality [204]. Kirillova et al. reported the 4D fabrication of hollow self-folding tubes composed of mouse bone marrow stromal cell-laden shape-morphing polymers (alginate and hyaluronic acid) with average internal tube diameters as low as 20 µm (similar to the size of small blood vessels) [205]. The self-folded tubes were able to support cell survival without influencing the cell viability for at least 7 days. Miao et al. reported the 3D laser printing of a smart and biocompatible porous scaffold composed of renewable soybean oil epoxidized acrylate and its ability to support human bone marrow mesenchymal stem cells (hMSCs) by increasing its adhesion and proliferation compared to polyethylene glycol diacrylate (PEGDA) [206]. Most importantly, the scaffold underwent temporary fixation at −18 °C and fully recovered its original shape at physiological temperature. Hendrikson et al. reported the 4D printing of two types of scaffolds (each having different pore network; 0/90° and 0/45°) composed of thermoresponsive shape memory polyurethane and seeded with human mesenchymal stromal cells (hMSCs) with the achievement of 100% cell viability when cultured in vitro for 14 days [207]. Particularly, it was found that both polyurethane scaffolds fully recovered to the permanent memorized shape once the temperature was heated above the transition temperature (T_trans_ = 32 °C); however, upon return to the original shape, the seeded cells underwent significant morphological changes. Jamal et al. reported the bio-origami method of photopatterning the fabrication of the self-folding of the curved poly (ethylene glycol) (PEG) hydrogels into anatomically sized structures [208]. Due to the difference in the molecular weight of PEG, the curved structure underwent differential swelling when exposed to aqueous solutions. Later, the cylindrical hydrogels composed of self-folded, photo-crosslinked PEG bilayers were seeded with insulin-secreting β-TC-6 cells and cell viability assessment tests showed a value of 90% over 8 weeks. Lin et al. reported the fabrication of a 4D-printed, patient-specific shape memory occlusion device used to potentially treat patients with atrial septal defects [209]. After implantation, this novel, biodegradable device composed of a shape memory (poly (lactic acid)) structure integrated with Fe_3_O_4_ magnetic particles was remotely controlled and the results show that the device was cellular-friendly and formed an endothelial cell layer. 

The common printing techniques that have been adopted for 4D fabrication are polyjet printing and syringe printing; however, other 3D printing techniques have the potential to take part in the future of 4D printing [210]. There is a crucial need to develop bioinks that are suitable for the 4D fabrication process especially in tissue engineering and regenerative medicine applications where the printed models are expected to be biocompatible and function in accordance with the dynamic physiological environment. Up until now, most of the smart, multi-functional 4D inks have been reported for non-biomedical applications, which ultimately leaves larger room for the search for novel, smart, multi-functional materials and self-assembling materials capable of acquiring a stable change in structure and function despite changes in stimuli or various stimuli [210]. 

## 8. Restrictions and Future Directions

Using 3D bioprinting as a biofabrication tool to create functional, anatomically accurate, clinically relevant-sized structures with the ability to grow and remodel is still in its early stage; however, there is a lot of potential growth in the field of paediatric CHD. One of the main obstacles facing scientists is the inability to attain physiological levels of cardiomyocytes in cell-laden printed structures prior to use in an in vivo setting [211]. With the incorporation of stem cells to the 3D-printed structures, more evidence suggests the ability of stem cells to trigger the recruitment of endogenous cardiomyocytes on-site via paracrine mechanisms. In parallel, there is also a lot of evidence to support the recruitment of endogenous cells in acellular tissue-engineered structures in preclinical settings, therefore questioning the need to encapsulate the 3D-printed structures with cells prior to implantation [183]. 

Despite the achievements in 3D bioprinting heart, blood vessels, heart valves, patches, and other scaffolds, none have made it to the clinics due to the limitations in acquiring biomaterials with comparable mechanical characteristics and biocompatibility with the native cardiac environment. Furthermore, the encapsulated cells must remain viable and perform all cellular functions when present in an in vivo setting. Due to the limitations of the synthetic and polymeric materials in terms of slow biodegradability, the absence of structural properties that biomimic the native cardiac environment, and the inability to grow and remodel with paediatric patients despite their enhanced biomechanical and biocompatibility properties, the 3D bioprinting of soft tissues could be sought out. With the current 3D bioprinters, it is difficult to print soft tissues that biomimic the native cardiac tissue and withstand surgical suturing and high pulse pressure when implanted in an in vivo setting due to their low mechanical properties. As such, FRESH printing could solve this problem by printing cell-laden structures composed of fragile, soft tissues such as collagen, hyaluronic acid, and decellularized extracellular matrix materials encapsulated by cells in air without collapsing [68]. Furthermore, those structures resemble the native extracellular matrix of the heart in terms of structural and biomechanical properties, which makes it ideal for encapsulated cells to grow and function in a healthy manner and further downstream allow for tissue growth and remodelling. This work is still in the proof-of-concept stage and has not been translated to clinical settings in treating paediatric patients with CHD, which leaves much room for potential growth in this area. Apart from the correct selection of biomaterials, longevity, and function of encapsulated cells, major technological advancements in 3D bioprinters must be achieved to print structures with a high resolution and accuracy that biomimic the native cardiac environment [211]. 

Through 3D bioprinting and 3D imaging, personalized medicine can be achieved where the patient’s own cells and biomaterials could be used to create patient-specific tissue-engineered structures to treat paediatric CHD. When coupled with computational modelling, 3D-bioprinted scaffolds can perform better in an in vivo environment due to the optimization process of the design of scaffolds as a result of the study of the local hemodynamic profile on-site [67]. By complementing experimental work with simulation of the 3D-printed scaffolds, a better prediction is reached on how the scaffolds will perform in the native cardiac environment. 

## 9. Conclusions

CHDs such as cyanotic heart disease, left-sided obstruction defects, and septation defects are treated through reconstructive heart surgery using non-living materials. The major challenge facing scientists, clinicians, and engineers in this field is the use of replacement materials in corrective heart surgeries that lack the potential to grow, remodel, or regenerate and could be the cause of a patient’s immunological rejection due to the foreign nature of the materials. Furthermore, other challenges are the inability to fabricate stable, functional, anatomically accurate-sized tissue structures within a realistic timeframe. Currently, prosthetic grafts, valves, conduits, and patches are made of those short-lived, degenerative materials, and as a result, multiple corrective cardiac surgeries are performed during the patient’s lifetime. Tissue engineering and stem cell therapy promise to change that by providing living materials for corrective heart surgery.

Three-dimensional bioprinting can be used as a tool for the creation of stable, durable, viable, and clinically sized grafts, patches, valves, and conduits composed of the three main constituents, stem cells, synthetic materials/biomaterials, and biomolecules in a predefined geometry, in highly controlled fashion and within a realistic timeframe, thereby allowing for enhanced function in physiological environments. By using this approach, the 3D-bioprinted tissue structures will be able to grow, remodel, and ultimately repair the abnormal cardiac tissue without the need for multiple surgical interventions. The selection of materials and the choice of stem cells and biomolecules used in the 3D bioprinting process should be chosen carefully according to the application. As such, the type of 3D bioprinting technique should also be chosen carefully according to the application and the unique combinations of the three main constituents. 

To allow for enhanced integration of the stable, functional bioprinted cardiovascular tissue constructs within the host cardiac environment, vascularization within the engineered cardiac tissue constructs is an important factor to consider due to the provision of the adequate supply of nutrients and gaseous exchange between the cells, thereby allowing cardiac cells to live, migrate, and proliferate. This will allow for the damaged cardiac tissue to regain its cardiac function and maintain homeostasis. Four-dimensional bioprinting is also a field worth exploring due to its high potential to produce cardiac tissue constructs suitable for treating congenital heart defects because of the manipulation of flexible, smart, and programmable materials that can dynamically change shape with time according to the change in stimuli. This is particularly crucial when present in in vivo conditions since the physiological environment is a dynamic environment that is constantly experiencing changes in stimuli. 

## Figures and Tables

**Figure 1 biomolecules-14-00861-f001:**
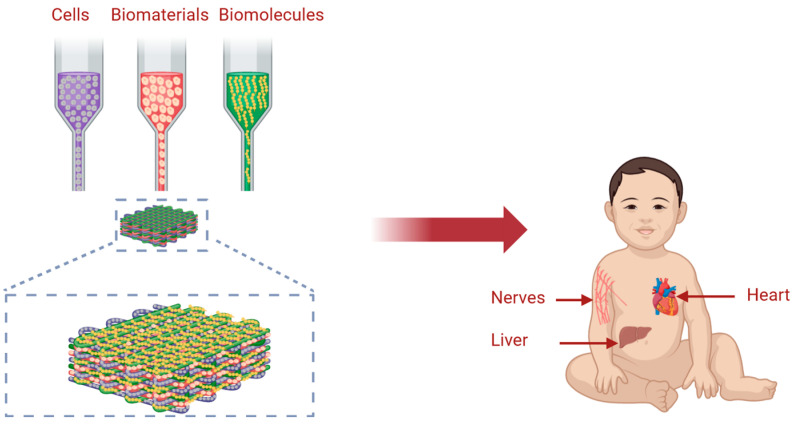
The main key players in the 3D bioprinting field are cells, biomaterials, and biomolecules that are spatially positioned in precise locations within the engineered constructs to fabricate complex 2D or 3D tissue/organ heterostructures (created with Biorender).

**Figure 2 biomolecules-14-00861-f002:**
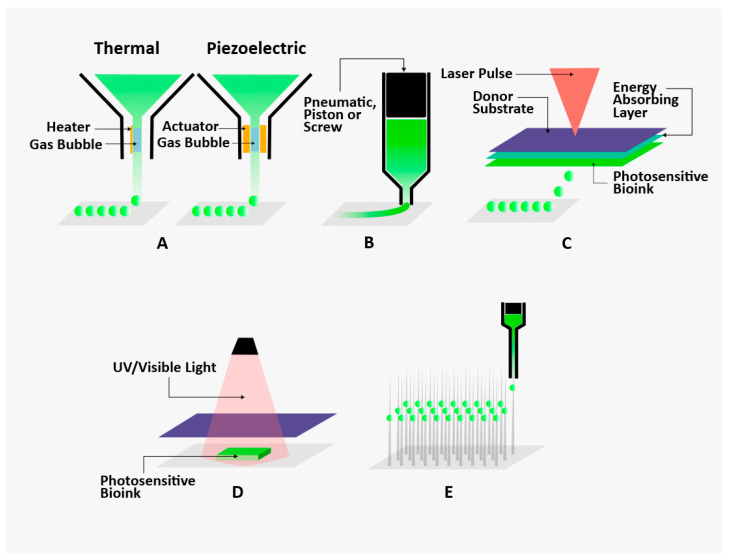
Schematic diagram of various 3D bioprinting technologies. (**A**) Inkjet bioprinting technology based on thermal and piezoelectric systems. (**B**) Robotic dispensing (extrusion-based) bioprinting technology based on pneumatic pressure, piston, and screw systems. (**C**) Laser-assisted bioprinting technology. (**D**) Stereolithography bioprinting technology including SLA and digital light processing (DLP) lased-based systems. (**E**) Scaffold-free bioprinting technology (created with Inkscape).

**Figure 3 biomolecules-14-00861-f003:**
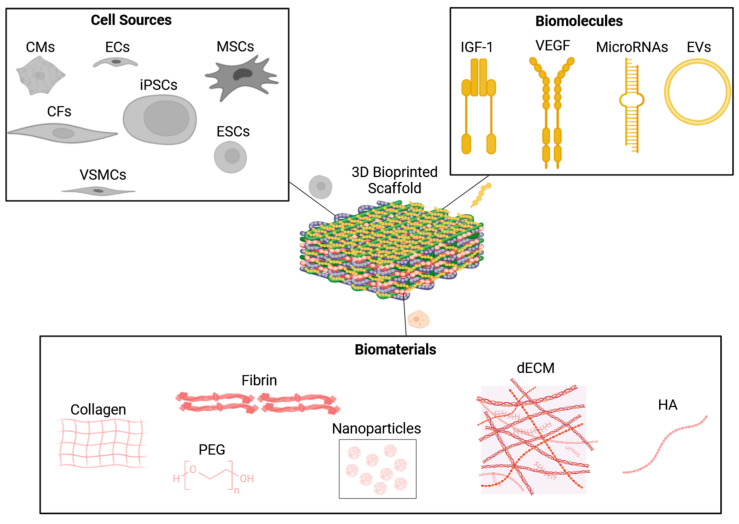
Various selections of cell sources, biomolecules, and biomaterials that make up a 3D-bioprinted structure. Each component is chosen based on the application with an aim to closely biomimic the native cardiac environment.

**Figure 4 biomolecules-14-00861-f004:**
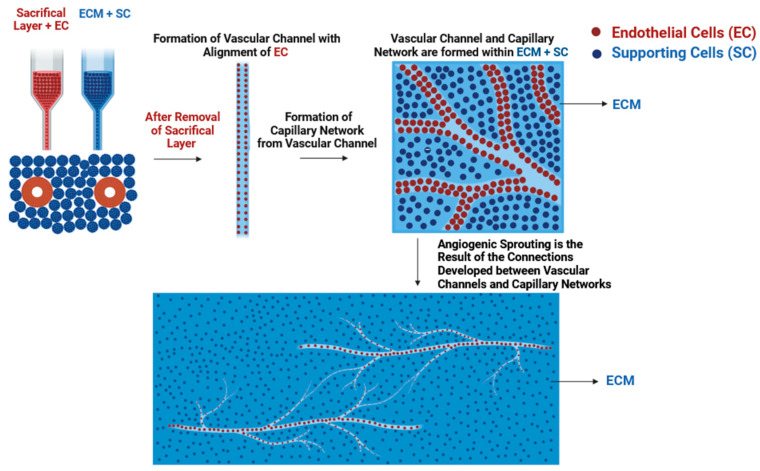
An illustration of the 3D bioprinting of vascularized tissue structures and supporting extracellular matrix (ECM) biomaterial encapsulated with supporting cells (SCs). After removal of the sacrificial layer, the encapsulated endothelial cells (ECs) form the lumen of vascular channels followed by the formation of capillary networks stemming out of the vascular channels within the ECM encapsulated with SCs. This results in angiogenic sprouting due to the connections developed between the vascular channels and capillary networks. The presence of supporting ECM biomaterial is needed to keep the vascularized tissue structures in place until tissue maturation is achieved.

**Table 1 biomolecules-14-00861-t001:** Summary of advantages and disadvantages of natural and synthetic biomaterials.

Type of Biomaterial	Advantages	Disadvantages
Natural Biomaterials	- Naturally occurring - Biocompatible - Non-toxic - Biodegradable (most natural biomaterials) - Non-immunogenic - Accessible - Ability to blend with other biomaterials to achieve superior characteristics	- Poor/unstable mechanical characteristics - Differences in batch-to-batch variation in source products
Synthetic Biomaterials	- Mechanically robust - Ability to blend with other biomaterials to achieve superior characteristics - Biocompatible - Printable - No batch-to-batch variation in source products	- Absence of protein-binding motifs that allow for cellular interactions/attachments

## Data Availability

Not applicable.

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
