# Peer review of "Recent Advances in Hydrogel-Based 3D Bioprinting and Its Potential Application in the Treatment of Congenital Heart Disease"

_biomolecules, 2024, doi:10.3390/biom14070861_

Round 1
Reviewer 1 Report
Comments and Suggestions for Authors
This review gives an overview of CHD and recent advancements in 3D bioprinting technologies with potential use in the treatment of CHD. The review is detailed and well organized. It will be more concise and valuable if parts of the content, such as printed materials, can be compared in the table.
The following two headings are the same and need to be modified.
2.1.4. Stereolithography (SLA)-Based 3D Bioprinting
2.1.5. Stereolithography (SLA)-Based 3D Bioprinting
Author Response
We thank the reviewer for these comments and suggestions. Below are our responses.
This review gives an overview of CHD and recent advancements in 3D bioprinting technologies with potential use in the treatment of CHD. The review is detailed and well organized. It will be more concise and valuable if parts of the content, such as printed materials, can be compared in the table.
Response: As suggested, a comparison table was added (page 26).
The following two headings are the same and need to be modified.
2.1.4. Stereolithography (SLA)-Based 3D Bioprinting
2.1.5. Stereolithography (SLA)-Based 3D Bioprinting
Response: This error was corrected (pages 19 & 20).
Reviewer 2 Report
Comments and Suggestions for Authors
This review gives an overview of CHD and recent advancements in 3D bioprinting technologies with potential use in the treatment of CHD. Moreover, the selection of appropriate biomaterials based on their chemical, physical and biological properties that are further manipulated to suit their application are also discussed. This review is well written. This review with abundant information and clear hierarchy, is a useful reference for the readers to understand this field. However, before publication, some concerns need addressing, primarily regarding the following aspects:
1. The authors focused on the hydrogel-based 3D bioprinting in the review. Thus, the title for this review can be specified.
2. More figures are suggested to be cited.
3. The advantages and disadvantages between natural biomaterials and synthetic biomaterials for 3D bioprinting scaffolds are required to be discussed in the end of Part 4.
Comments on the Quality of English Language
The Quality of English Language is good as it is.
Author Response
We thank the reviewer for these comments and suggestions. Below are our responses.
This review gives an overview of CHD and recent advancements in 3D bioprinting technologies with potential use in the treatment of CHD. Moreover, the selection of appropriate biomaterials based on their chemical, physical and biological properties that are further manipulated to suit their application are also discussed. This review is well written. This review with abundant information and clear hierarchy, is a useful reference for the readers to understand this field. However, before publication, some concerns need addressing, primarily regarding the following aspects:
- The authors focused on the hydrogel-based 3D bioprinting in the review. Thus, the title for this review can be specified.
Response: As suggested, the title was altered to be more specific (page 1).
- More figures are suggested to be cited.
Response: As suggested, two more figures (3 & 4) and a table were added (pages 26, 30 and 32).
- The advantages and disadvantages between natural biomaterials and synthetic biomaterials for 3D bioprinting scaffolds are required to be discussed in the end of Part 4.
Response: As requested, a comparison between natural and synthetic biomaterials was added (pages 25 & 26).
Reviewer 3 Report
Comments and Suggestions for Authors
The paper provides an interesting explanation of CHD, as well as a detailed summary of the materials that can be used to treat it.
It has some concerns:
- - A paragraph is required at the end of the Introduction section, in which the main novelty of the paper is addressed. In addition, a summary of the content of the different sections is recommended.
- - Section 2.1. When introducing the different bioprinting technologies, for better understanding, I recommend to use the classification and nomenclature defined by ISO/ASTM 52900 standard, which defines 7 groups of technologies: VAT photopolymerization, material extrusion, binder jetting, powder bed fusion, material jetting, sheet lamination and directed energy deposition.
- - Section 3.2. Lines 624 and 625. The explanation about the 3D printing parameters should be extended, in order to mention most important parameters such as infill rate, infill pattern, etc.
- - Section 3.6 about biocompatibility should be extended in order to include some examples of biocompatible materials that are used for the treatment of CHD. For example, 3D printed zirconia parts have been proved to be biocompatible.
https://journals.sagepub.com/doi/10.1177/09544054231168469
- - Section 6. I recommend to add more references regarding extrusion processes, for example about the use of FDM extrusion techniques to obtain vascular stents.
https://www.mdpi.com/1996-1944/11/9/1679
- - In Section 6, if possible, I recommend to add some pictures about the examples of the 3D printed structures.
- - The conclusions should be expanded in order to include the previous work about 3D printing of medical devices that are related to heart diseases.
Other minor concerns are as follows:
- In the title, “herat” should be “heart”.
- In line 260, pL should be pl.
- The title of section 2.1.5 seems to be wrong.
- Line 1415. A dot is expected before the word “therefore”.
Author Response
We thank the reviewer for these comments and suggestions. Below are our responses.
The paper provides an interesting explanation of CHD, as well as a detailed summary of the materials that can be used to treat it.
It has some concerns:
- - A paragraph is required at the end of the Introduction section, in which the main novelty of the paper is addressed. In addition, a summary of the content of the different sections is recommended.
Response: As requested, a paragraph was added at the end of the introduction (page 4).
- - Section 2.1. When introducing the different bioprinting technologies, for better understanding, I recommend to use the classification and nomenclature defined by ISO/ASTM 52900 standard, which defines 7 groups of technologies: VAT photopolymerization, material extrusion, binder jetting, powder bed fusion, material jetting, sheet lamination and directed energy deposition.
Response: The text in this section was altered to use the recommended classification and nomenclature (page 6).
- - Section 3.2. Lines 624 and 625. The explanation about the 3D printing parameters should be extended, in order to mention most important parameters such as infill rate, infill pattern, etc.
Response: The suggested explanation was added to the section about bio-printability (page 14).
- - Section 3.6 about biocompatibility should be extended in order to include some examples of biocompatible materials that are used for the treatment of CHD. For example, 3D printed zirconia parts have been proved to be biocompatible.
https://journals.sagepub.com/doi/10.1177/09544054231168469
Response: As suggested, an example was added (page 16).
- - Section 6. I recommend to add more references regarding extrusion processes, for example about the use of FDM extrusion techniques to obtain vascular stents.
https://www.mdpi.com/1996-1944/11/9/1679
Response: As suggested, more references were added (pages 31-32).
- - In Section 6, if possible, I recommend to add some pictures about the examples of the 3D printed structures.
Response: As suggested, a figure was added (page 32).
- - The conclusions should be expanded in order to include the previous work about 3D printing of medical devices that are related to heart diseases.
Response: Although previous work about 3D printing (without cells) of medical devices related to heart diseases is very interesting, we believe that it is beyond the scope of this review which is more focused on 3D bioprinting. Particularly, the suggested addition would not fit under the new more specific (hydrogel-based 3D bioprinting) title of this manuscript suggested by one of the reviewers.
Other minor concerns are as follows:
- In the title, “herat” should be “heart”.
- In line 260, pL should be pl.
- The title of section 2.1.5 seems to be wrong.
- Line 1415. A dot is expected before the word “therefore”.
Response: All these minor errors were corrected.
Round 2
Reviewer 3 Report
Comments and Suggestions for Authors
The paper has improved significantly.
In the introduction, a final sentence should be added that summarizes the content and the organization of the paper.
Author Response
Comment: In the introduction, a final sentence should be added that summarizes the content and the organization of the paper.
Response: As suggested, a sentence was added.